# Crisp: A Spectral-Based Interaction Strategy for Multivariate Time Series Forecasting

**Binwu Wang** [1]  **Gaoyun Lin** [1]  **Jiaming Ma** [1]  **Qihe Huang** [1]  **Zhengyang Zhou** [1]
**Xu Wang** [1]  **Pengkun Wang** [1] [*]  **Yang Wang** [1] [*]

## Abstract

Multivariate time series (MTS) forecasting critically relies on effectively modeling inter-variable dependencies. However, existing paradigms often face an inherent trade-off: channel-isolation strategies may lead to information fragmentation in strongly coupled systems, while channel-interaction methods can introduce spurious dependencies among irrelevant variables. To address this challenge, we propose **C**oherent **R**esonance **I**nteraction with **S**pectral **P**riors (Crisp), a novel framework built on the principle that effective information exchange should occur only among variables exhibiting compatible oscillatory patterns. Specifically, Crisp derives spectral priors in the frequency domain to construct dynamic resonance topologies. Through a differentiable, adaptive, and strictly sparse blocking mechanism, Crisp explicitly sets the attention weights of spectrally inconsistent neighbors to zero, thereby suppressing spurious interactions. Furthermore, we introduce a spectral-gated feature filtering module that refines variable representations according to their intrinsic spectral characteristics. Extensive experiments demonstrate that Crisp achieves the best or highly competitive performance across most settings. The code of Crisp is available at [GitHub](GitHub).

## 1. Introduction

Time series forecasting plays a critical role in data-driven applications such as financial analysis, traffic flow management, and weather prediction (Jin et al., 2024; Liu et al., 2024; 2025; Ma et al., 2025; Wang et al., 2024b).

---

[*]Corresponding authors.  [1]University of Science and Technology of China, Hefei, China. Correspondence to: Binwu Wang <wbw2024@ustc.edu.cn>.

*Proceedings of the $43^{rd}$ International Conference on Machine Learning*, Seoul, South Korea. PMLR 306, 2026. Copyright 2026 by the author(s).

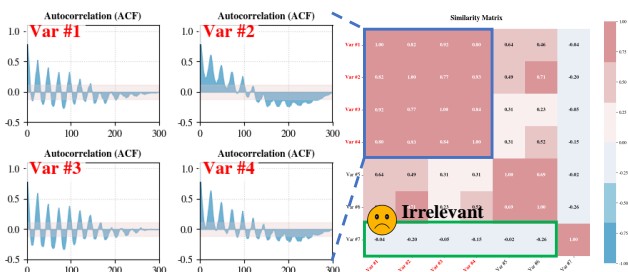

*Figure 1.* For the ETTh1 dataset, the right panel illustrates the Pearson correlation coefficients among variables. It can be observed that Variables #1 to #4 exhibit clear positive correlations. However, the left panel presents the autocorrelation functions of Variables #1 to #4, revealing their underlying and potentially distinct temporal trends.

Many real-world forecasting problems are inherently multivariate, involving multiple correlated variables that coexist and jointly shape future dynamics. Therefore, forecasting models are expected to effectively capture and exploit the dynamic latent dependencies among variables as they evolve over time. Existing methods for modeling inter-variable dependencies can be broadly categorized into two paradigms. ❶ **Channel-Isolation Strategy** treats multivariate time series as a collection of independent univariate channels and explicitly prevents information exchange across variables (Nie et al., 2023; Shao et al., 2022). Although simple and effective in certain scenarios, this strategy inevitably leads to inefficient utilization of cross-variable information and imposes inherent performance limitations. In contrast, ❷ **Channel-Interaction Strategy** encourages information exchange along the variable dimension. Specifically, some methods assume the presence of global inter-variable dependencies, resulting in channel-dependence modeling strategies (Zhang & Yan, 2023; Liu et al., 2024; Hu et al., 2025a). Beyond this, several recent clustering-based approaches (Qiu et al., 2025; Chen et al., 2024; Sun et al., 2025; Hu et al., 2025b) further refine this paradigm by employing KNN or clustering techniques to group variables based on observed similarities, and then restricting interactions within each local group. This design reduces redundant information exchange while preserving meaningful local dependencies.

However, existing strategies still suffer from significant lim-

itations. Channel-isolation-based methods ignore joint information among variables and often lead to suboptimal performance (Shao et al., 2024). Advanced channel-interaction strategies typically rely on self-attention mechanisms to compute inter-variable dependencies, which can introduce redundant and noisy interactions. In particular, for variables that are weakly correlated with others, such as Variable #7 in Figure 1, the Softmax function in self-attention assigns nonzero weights even to low-similarity interactions, thereby injecting noise into the learned representations. Although clustering-based methods can restrict the scope of inter-variable interactions, they are prone to assigning heterogeneous variables to so-called pseudo-homogeneous groups. As shown in Figure 1, Variables #1 to #4 exhibit relatively high temporal similarity, yet their underlying trends differ substantially. Interactions among these variables may therefore propagate misleading information and degrade overall performance. Finally, existing mechanisms struggle to effectively isolate missing-value interference, often misinterpreting missing patterns as meaningful structures. Such errors can further propagate noise across variables through interaction mechanisms, severely undermining the robustness of the overall representation.

In this paper, we propose **C**oherent **R**esonance **I**nteraction with **S**pectral **P**riors (Crisp), which introduces a more precise strategy for multivariate time series forecasting. Our core principle is that effective information exchange should occur only between variables with compatible oscillatory patterns, namely those that are spectrally consistent. To this end, we first adopt a dual-stream heterogeneous encoder to disentangle long-term components. Crisp then extracts spectral priors to characterize the intrinsic oscillatory properties of each variable. Based on these priors, we construct a dynamic resonance topology, which serves as an interaction bias to prune interactions between spectrally incompatible variables. This design enables precise selective interaction, effectively decoupling spectrally inconsistent neighboring variables and improving robustness to missing values, as illustrated in Figure 2. In addition, we introduce a spectrum-gated feature filtering module to further refine the representations.

Our **contributions** are as follows:

- **Perspective**: We propose Crisp, which establishes a flexible interaction paradigm that enables robust joint modeling of variable dependencies while strictly preventing the emergence of spurious and pseudo-interactions.

- **Methodology:** We incorporate a Frequency-Adaptive Selective Interaction module that utilizes $\alpha$-Entmax to rigorously force the attention weights of non-resonating variables to be exactly zero, complemented by a Spectrum-Gated Feature Filtration module. We further explain the

effectiveness of the design from a gradient perspective.

- **Performance:** Extensive experiments on real-world datasets demonstrate that Crisp outperforms existing time series forecasting baselines, achieving state-of-the-art performance on over **90%** of the evaluated metrics.

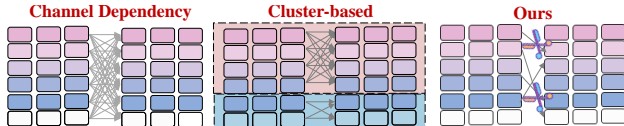

*Figure 2.* An illustration of three channel-interaction strategies, where our method can block pseudo-correlated interactions.

## 2. Related Work

**Time Series Forecasting**     Time series forecasting has been widely applied across numerous domains (Wang et al., 2023b; Huang et al., 2023b; Wang et al., 2024a). Recent models often focus on capturing intra-variable patterns by pairing Transformer- (Ma et al., 2026b;a) or MLP-based (Huang et al., 2026; Xia et al., 2026) backbones with representations that emphasize periodic structures across multiple temporal scales. For instance, Autoformer (Wu et al., 2021b) and Fedformer (Zhou et al., 2022) integrate series decomposition and frequency-domain representations into attention to better model multi-scale temporal structures. Other approaches further treat periodicity as a core design principle (Wu et al., 2023b; Lin et al., 2024). Beyond temporal dynamics, a central challenge in multivariate forecasting is modeling inter-variable dependencies. More recently, driven by the rapid progress of large language models, researchers have begun to incorporate LLMs into time series forecasting (Jin et al., 2023), leading to the development of various time-series foundation models (Liang et al., 2024; Goswami et al., 2024). Meanwhile, the integration of multimodal information, such as text and images, has also emerged as a promising direction for enhancing forecasting performance (Jiang et al., 2025; Jia et al., 2026). Accordingly, existing methods can be broadly grouped into channel-isolation and channel-interaction strategies, depending on whether cross-variable interactions are explicitly modeled.

**Channel-isolation Strategy in TSF**     Some methods adopt channel isolation (independence) strategies (Nie et al., 2022; Shao et al., 2022; Wang et al., 2025a; Das et al., 2023), which process each time series channel (variable) separately, ignoring any dependencies or interactions across channels. Intuitively, this simplistic approach may fail to achieve optimal performance in many cases, as it does not account for the potentially complex relationships inherent in multivariate time series data. Representative models include PatchTST (Nie et al., 2022), FITS (Xu et al., 2023b), Time-Former (Liu et al., 2025), and DLinear (Zeng et al., 2023).

These approaches have been widely used in recent years. However, the lack of explicit inter-variable dependency modeling has become a bottleneck, limiting their performance in multivariate settings.

**Channel-interaction Strategy in TSF** In contrast, channel-interaction strategies jointly model temporal dynamics and inter-variable relationships. Representative architectures include graph neural networks (Wang et al., 2023a; Zhou et al., 2023), convolutional neural networks (Wang et al., 2023c), and MLP-based models (Huang et al., 2024). Among them, attention-based solutions are particularly prominent, as they can dynamically capture latent dependencies among variables and thereby better characterize multivariate relationships. However, these strategies often induce dense interactions, introducing a large number of redundant or invalid connections and consequently degrading model robustness. To mitigate this issue, several recent studies (e.g., (Chen et al., 2024; Qiu et al., 2025; Wang et al., 2025b)) have proposed channel-clustering paradigms based on variable similarity, adapting the modeling strategy according to inter-variable similarity. For instance, Time-Filter (Hu et al., 2025b) employs KNN to compute dynamic similarities among variables. TQNet (Lin et al., 2025) uses learnable coarse-grained cached query vectors, which essentially achieve adaptive clustering via attention mechanisms. TFPS (Sun et al., 2025) introduces subspace clustering and incorporates a mixture of experts mechanism for modeling.

Despite these advances, existing strategies remain inadequate for complex multivariate forecasting systems. Clustering-based and graph-based methods may still retain unnecessary interactions in complex multivariate settings, which ultimately degrades model robustness.

## 3. Problem Formulation

Let $\mathbf{x}_t \in \mathbb{R}^N$ denote the observed multivariate time series at discrete time index $t$, where $N$ is the number of variables. Given the input $\mathbf{X} = [\mathbf{x}_{t-T+1}, \cdots, \mathbf{x}_t] \in \mathbb{R}^{T \times N}$ with the look-back window $T$, our objective is to forecast the values over the next $P$ time steps, denoted by $\mathbf{Y} = [\mathbf{x}_{t+1}, \cdots, \mathbf{x}_{t+P}] \in \mathbb{R}^{P \times N}$.

## 4. Methodology

### 4.1. Overview of Crisp Architecture

As illustrated in Figure 3 and Algorithm 1, Crisp employs a dual-stream inverted decoupling architecture. To mitigate the inherent non-stationarity of multivariate time series, we first apply Reversible Instance Normalization (RevIN) (Kim et al., 2021) to standardize the input. The normalized input $\mathbf{X}$ is then decomposed into two orthogonal components using a moving-average kernel:

■ **Trend Component ($\hat{\mathbf{X}}$):** Represents long-term progression, modeled via a linear regressor: $\hat{\mathbf{Y}}_t = \text{Linear}(\hat{\mathbf{X}})$.

■ **Seasonal Component ($\overline{\mathbf{X}}$):** Contains high-frequency variations and complex dependencies. This component is processed by the core Crisp encoder to yield $\hat{\mathbf{Y}}_s$.

The final prediction is the sum of these branches: $\hat{\mathbf{Y}} = \hat{\mathbf{Y}}_{trend} + \hat{\mathbf{Y}}_{sea}$, which is then mapped back to the original distribution via RevIN.

### 4.2. Dual-stream Heterogeneous Decoupling Encoder

Beyond temporal decoupling, to accurately capture the complex evolutionary dynamics of multivariate time series, we introduce a temporal dynamics encoder. This module is designed to extract multi-granularity temporal features while constraining the learned representations using important spatiotemporal prior information.

**Multi-Scale Feature Encoding.** Given the normalized input sequence for the $i$-th variate $x^{(i)} \in \mathbb{R}^T$ in $\overline{\mathbf{X}}$, we employ a parallel multi-branch architecture to encode patterns with varying receptive fields. (1) Global & Pointwise Dynamics: We utilize a linear projection to encode the holistic trend, alongside a $1 \times 1$ convolution to preserve instantaneous signal fidelity without aggregation. (2) Local Multi-Scale Dynamics: To capture local periodic patterns, we employ a set of 1D convolutions with diverse kernel sizes $\mathcal{K} \in \{3, 5, 7\}$. Formally, the feature extraction process is defined as:

$$
\begin{aligned}
\mathbf{h}_g^{(i)} &= \mathbf{W}_g x^{(i)}, \\
\mathbf{h}_1^{(i)} &= \text{Leaky\_relu}(\text{Conv1d}_{1 \times 1}(x^{(i)})), \\
\mathbf{h}_k^{(i)} &= \text{Pool}(\text{Leaky\_relu}(\text{Conv1d}_k(x^{(i)}))), \quad k \in \mathcal{K}
\end{aligned} \tag{1}
$$

where $\text{Pool}(\cdot)$ represents the pooling operation. The multi-scale shape encoding $\mathbf{E}_s^{(i)}$ is obtained by fusing these branches via a projection layer $\mathbf{W}_f$.

**Temporal Prior Information Encoding.** To associate the encoded dynamic information with absolute time, we map hierarchical timestamps, such as the hour of the day and the day of the week, into learnable vectors $\mathbf{E}_t$. These external temporal context cues help identify common periodic patterns in data evolution (e.g., traffic peak periods) (Shao et al., 2022; Ma et al., 2025).

**Learnable Variable Conditioning.** We assign each variable a distinct learnable embedding vector to explicitly encode variable identity, denoted by $\mathbf{E}_v$, enabling the model to learn variable-specific characteristics.

The final encoded representation $\overline{\mathbf{H}}$ is the superposition of the dynamic shape features and static priors:

$$
\overline{\mathbf{H}} = \mathbf{E}_s + \mathbf{E}_t + \mathbf{E}_v \in \mathbb{R}^{T \times N \times D} \tag{2}
$$

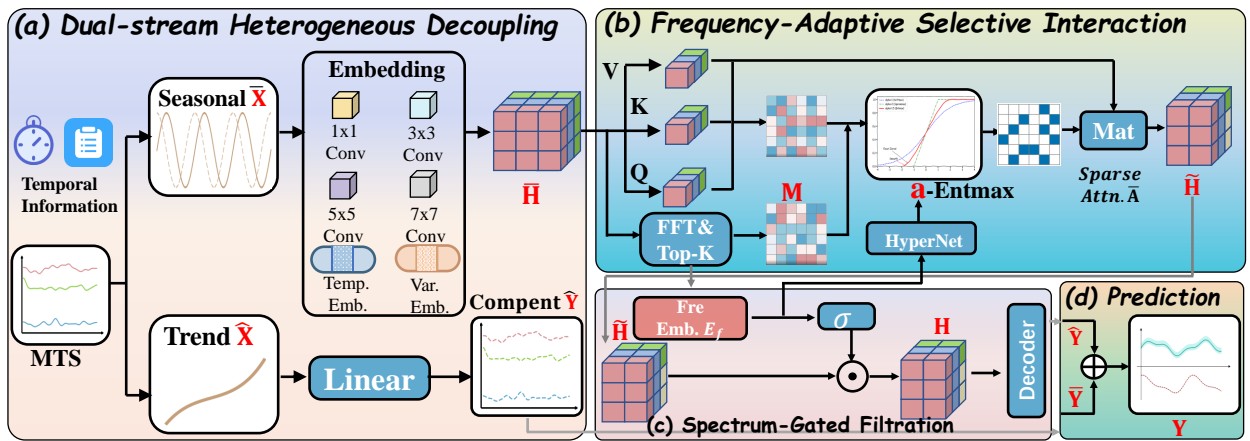

*Figure 3.* The overall architecture of the model.

## 4.3. Frequency-Adaptive Selective Interaction

Channel-interaction strategies mix variables to capture inter-variable dependencies, but in weakly correlated or heterogeneous environments they tend to introduce irrelevant signals, leading to noise interference and performance degradation. A typical architecture employs the Transformer to compute similarities among variables (Zhang & Yan, 2023; Qiu et al., 2025; Liu et al., 2024):

$$\mathbf{A} = \text{Softmax}\left(\frac{\mathbf{Q}\mathbf{K}^\top}{\sqrt{D}}\right) \in \mathbb{R}^{N \times N},$$
$$\mathbf{Q} = \mathbf{W}_q \overline{\mathbf{H}}, \quad \mathbf{K} = \mathbf{W}_k \overline{\mathbf{H}}, \quad \mathbf{V} = \mathbf{W}_v \overline{\mathbf{H}}, \quad (3)$$

Modeling variable correlations within Transformer architectures faces two fundamental challenges: ❶ Correlation measurements based on pointwise proximity are inherently sensitive to noise and spurious correlations in the time domain. ❷ The standard Softmax function lacks the ability to strictly filter out irrelevant dynamics, as it assigns nonzero attention weights even to completely unrelated variables.

We propose a frequency-adaptive selective interaction mechanism to improve this process. Specifically, we leverage spectral coherence to dynamically identify the compatibility of evolutionary rhythms and allow information exchange only between variables that exhibit coherent frequency dynamics, while filtering out unnecessary interactions.

### 4.3.1. QUANTIFYING SPECTRAL COHERENCE

We posit that valid interactions should be grounded in physical consistency. To capture this, we perform Fast Fourier Transform (FFT) on the seasonal representations to obtain the amplitude spectrum $\mathcal{Z} \in \mathbb{R}^{N \times F}$. To ensure robustness against high-frequency stochastic fluctuations, we apply a low-pass filter, retaining only the $K\%$ discriminative components, denoted as $\mathcal{Z}'$. A **Dynamic Resonance Topology**

$\mathbf{M}$ is then derived via cosine similarity:

$$\mathbf{M}_{i,j} = \text{CosSim}(\text{MLP}(\mathcal{Z}'_i), \text{MLP}(\mathcal{Z}'_j)) \quad (4)$$

Here, $\mathbf{M}_{i,j}$ serves as a coherence metric: a high value implies that variables $i$ and $j$ resonate at similar frequencies, validating the necessity of information exchange.

### 4.3.2. ADAPTIVE STRICT-BLOCKING MECHANISM

$\alpha$-**Entmax Transformer.** We use $\mathbf{M}$ to modulate the attention generation process and implement a selective interaction strategy:

$$\mathbf{S} = \frac{\mathbf{Q}\mathbf{K}^\top}{\sqrt{D}} + \lambda \cdot \mathbf{M} \in \mathbb{R}^{N \times N}, \quad (5)$$

where $\lambda$ is a learnable parameter used to control the bias fusion weight. We then apply the $\alpha$-Entmax activation function (Peters et al., 2019). Unlike standard Softmax, $\alpha$-Entmax provides strict noise-suppression behavior by mapping weights below a dynamically learned threshold to exactly zero. Formally, the attention weights are derived by maximizing the Tsallis entropy family. The closed-form solution of $\alpha$-Entmax corresponds to a thresholded projection:

$$\overline{\mathbf{A}}_{ij} = [(\alpha - 1)(\mathbf{S}_{ij} - \tau)]_+^{\frac{1}{\alpha-1}} \quad (6)$$

where $[x]_+ = \max(x, 0)$ denotes the rectification function, and $\tau$ is a Lagrangian multiplier ensuring $\sum_j \overline{\mathbf{A}}_{ij} = 1$. Whenever the interaction score $\mathbf{S}_{ij}$ falls below the dynamic threshold $\tau$, the term inside the bracket becomes negative, and the rectification operation strictly forces the weight $\overline{\mathbf{A}}_{ij}$ to be exactly zero. This provides a mathematical guarantee of complete filtering of irrelevant signals in Section 4.4. The final input representation is obtained by multiplying the attention matrix $\overline{\mathbf{A}}$ with the value vector $\mathbf{V}$, denoted as $\widetilde{\mathbf{H}}$.

**Spectral-Context HyperNetwork for $\alpha$ Learning.** $\alpha$ is not a fixed scalar but is adaptively inferred from the global frequency context of the multivariate system. Specifically, we

first aggregate the frequency embeddings $\mathbf{E}_f = \text{MLP}(\mathcal{Z}') \in \mathbb{R}^{N \times D}$ via global average pooling to obtain a system-level spectral representation $\mathbf{g} \in \mathbb{R}^D$:

$$\mathbf{g} = \frac{1}{N} \sum_{i=1}^{N} \mathbf{E}_f^{(i)} \qquad (7)$$

This global vector $\mathbf{g}$ is then fed into an MLP layer to regress the optimal sparsity level:

$$\alpha = 1 + \sigma\left(\mathbf{W}_2 \tanh(\mathbf{W}_1 \mathbf{g} + \mathbf{b}_1) + \mathbf{b}_2\right) \in (1, 2) \quad (8)$$

where $\mathbf{W}_1, \mathbf{W}_2, \mathbf{b}_1$, and $\mathbf{b}_2$ are learnable weights, and $\sigma$ is the Sigmoid function.

### 4.3.3. SPECTRUM-GATED FEATURE FILTRATION

To further refine inter-variable interactions, we propose a spectral-gated feature filtering module. Acting as a post-interaction refinement filter, it leverages previously extracted frequency embeddings to selectively suppress irrelevant feature channels, formulated as follows:

$$\mathbf{H} = \text{GELU}(\mathbf{W}_c \widetilde{\mathbf{H}}) \odot \sigma(\mathbf{W}_g \mathbf{E}_f) \qquad (9)$$

where $\mathbf{W}_c$ and $\mathbf{W}_g$ are learnable parameters. This ensures that the feature transformation is contextually adapted to the dominant frequency components of each specific variable. Finally, $\mathbf{H}$ is fed into a decoder (e.g., a single fully connected layer) to produce the prediction outputs $\hat{\mathbf{Y}}_s$.

### 4.4. Theory Analysis

**Spectrally Structured Gradient Blocking.** We clarify that the zero-gradient behavior discussed below is not a newly discovered property of $\alpha$-Entmax itself. It follows from the known sparse support of $\alpha$-Entmax (Peters et al., 2019). The purpose of this analysis is instead to characterize how this property is instantiated in Crisp, where the interaction support is jointly determined by the learned attention affinity and the spectral resonance topology $\mathbf{M}$:

$$\mathbf{S}_{ij} = \frac{\mathbf{Q}_i \mathbf{K}_j^\top}{\sqrt{D}} + \lambda \mathbf{M}_{ij}. \qquad (10)$$

Thus, the active neighbor set is selected not by $\alpha$-Entmax in isolation but by applying $\alpha$-Entmax to the spectrally biased scores:

$$\mathcal{V}_i = \{j \mid \overline{\mathbf{A}}_{ij} > 0\} = \{j \mid \mathbf{S}_{ij} > \tau_i\}. \qquad (11)$$

Here $\tau_i$ is the normalization threshold for row $i$. Since $\mathbf{M}_{ij}$ measures frequency-domain compatibility, the spectral topology explicitly participates in whether a variable pair crosses the threshold and enters the active support. In this sense, Crisp turns the generic sparsity of $\alpha$-Entmax into a spectral-topology-guided interaction rule.

Let $\mathcal{L}$ denote the objective function. The gradient of the loss with respect to $\mathbf{S}_{ij}$ is given by

$$\frac{\partial \mathcal{L}}{\partial \mathbf{S}_{ij}} = \sum_{k=1}^{N} \frac{\partial \mathcal{L}}{\partial \overline{\mathbf{A}}_{ik}} \frac{\partial \overline{\mathbf{A}}_{ik}}{\partial \mathbf{S}_{ij}}. \qquad (12)$$

For $\alpha$-Entmax, the Jacobian is supported only on active entries. Let $Z_i = \sum_{m \in \mathcal{V}_i} \overline{\mathbf{A}}_{im}^{2-\alpha}$. Then

$$\frac{\partial \overline{\mathbf{A}}_{ik}}{\partial \mathbf{S}_{ij}} = \begin{cases} \overline{\mathbf{A}}_{ik}^{2-\alpha} \left( \delta_{kj} - \frac{\overline{\mathbf{A}}_{ij}^{2-\alpha}}{Z_i} \right), & k, j \in \mathcal{V}_i, \\ 0, & \text{otherwise.} \end{cases} \qquad (13)$$

Consequently, if a neighbor $j$ is excluded by the Crisp interaction mechanism, then

$$\begin{aligned} \frac{\partial \mathcal{L}}{\partial \mathbf{S}_{ij}} &= 0, \\ \frac{\partial \mathcal{L}}{\partial \mathbf{M}_{ij}} &= \lambda \frac{\partial \mathcal{L}}{\partial \mathbf{S}_{ij}} = 0, \end{aligned} \qquad \forall j \notin \mathcal{V}_i. \qquad (14)$$

The theoretical contribution of this section is therefore not a restatement of the fact that $\alpha$-Entmax can produce zero weights. Rather, it explains the mechanism by which Crisp aligns forward sparsity and backward gradient flow with spectral consistency: non-resonant neighbors are removed from the attention support, and their spectral edges receive no back-propagated update through the interaction pathway. This provides a gradient-based interpretation of why the proposed spectral topology helps prevent irrelevant variables from influencing the representation of variable $i$.

## 5. Experiments

In this section, we evaluate Crisp by answering the following potential research questions: **(Q1)** Model performance; **(Q2)** ablation studies; **(Q3)** efficiency; **(Q4)** robustness for missing values; **(Q5)** analysis of key hyperparameters, and **(Q6)** visualization of model attention weights.

### 5.1. Experimental Setup

**Datasets** We evaluate the proposed method on 10 widely used real-world datasets, including the ETT series (Zhou et al., 2021), Electricity, Solar, Weather (Wu et al., 2021a), AQShunyi, AQWan, CzeLan, and ZafNoo (Qiu et al., 2025). For all datasets, the look-back window $T$ is set to 96. The future horizon $P$ is fixed to $\{12,24,48\}$ for the METR-LA dataset to evaluate short-term forecasting performance, and to $\{96,192,336,720\}$ for the other datasets to assess long-term forecasting performance. These datasets exhibit substantial diversity in terms of scale, number of variables, and application domains. Detailed statistics of the datasets are summarized in Table 1.

*Table 1.* Detailed information about the datasets.

| Dataset | #Dim | Timesteps | Split | Frequency | Domain |
|---|---|---|---|---|---|
| ETTh1 | 7 | 14,400 | 6:2:2 | 1 hour | Electricity |
| ETTh2 | 7 | 14,400 | 6:2:2 | 1 hour | Electricity |
| ETTm1 | 7 | 57,600 | 6:2:2 | 15 mins | Electricity |
| ETTm2 | 7 | 57,600 | 6:2:2 | 15 mins | Electricity |
| Electricity | 321 | 26,304 | 7:1:2 | 1 hour | Electricity |
| Solar | 137 | 52,560 | 7:1:2 | 10 mins | Energy |
| METR-LA | 207 | 34,272 | 7:1:2 | 5 mins | Transportation |
| AQWan | 11 | 35,064 | 7:1:2 | 1 hour | Environment |
| CzeLan | 11 | 19,934 | 7:1:2 | 30 mins | Nature |
| ZafNoo | 11 | 19,225 | 7:1:2 | 30 mins | Nature |

**Protocol Settings** All experiments are implemented in PyTorch and executed on an NVIDIA GeForce RTX 4090D GPU with 24 GB of memory. For our model, we use the AdamW optimizer (Kingma, 2014) for optimization and adopt mean absolute error (MAE) as the training loss, with mean squared error (MSE) and MAE as the primary evaluation metrics. We disable the drop-last trick. For most datasets, the learning rate is selected from { 0.0001 , 0.0005}, and the model dimensionality is tuned from {64, 128, 256} to find the optimal setting. For additional details on hyperparameters and settings, please refer to Appendix.

**Baselines** To assess the performance of our model, we compare it with advanced time series forecasting models from recent years. These include DLinear (Zeng et al., 2023), TimesNet (Wu et al., 2023a), PatchTST (Nie et al., 2023), iTransformer (Liu et al., 2024), CrossGNN (Huang et al., 2023a), SOFTS (Han et al., 2024), Crossformer (Zhang & Yan, 2023), FilterNet (Yi et al., 2024), FITS (Xu et al., 2023a), CycleNet (Lin et al., 2024), FreTS (Yi et al., 2023), TimeMixer (Wang et al., 2024c), Duet (Qiu et al., 2025), xPatch(Stitsyuk & Choi, 2025), MultiPatchFormer (Naghashi et al., 2025), WPMixer (Murad et al., 2025), TQNet (Lin et al., 2025), SimpleTM (Chen et al., 2025), PatchMLP (Tang & Zhang, 2025), TimeKAN (Huang et al., 2025), TimeFilter (Hu et al., 2025b), and TFPS (Sun et al., 2025). We group the baselines according to their interaction mechanisms, and the complete list of baselines is provided in **Appendix C.1**.

### 5.2. Overall Performance (Q1)

Tables 2 and 3 present comparisons between Crisp and baseline models for long-term forecasting, while Figure 4 shows its performance against several advanced models on short-term forecasting tasks. In these evaluations, lower MSE and MAE values indicate higher predictive accuracy.

Channel-independent models such as PatchTST and DLinear lack explicit mechanisms for modeling inter-variable dependencies, which limits their performance in MTS. iTransformer achieves relatively low error rates; however, because it relies on Transformer-based mechanisms to model inter-variable dependencies, it is prone to introducing noise, which leaves room for further performance improvement. DUET and TimeFilter cluster variables based on observed similarities to constrain inter-variable interactions, while TQNet adopts a cached-Q Transformer variant to extract global temporal patterns, where the Q vectors are coarse-grained and implicitly perform aggregation. These strategies help avoid redundant interactions and thus achieve competitive performance. For short-term forecasting, accurately modeling inter-variable relationships becomes increasingly important, particularly in strongly coupled systems such as traffic networks. DUET performs poorly. One plausible explanation is that DUET's coarse-grained clustering may group spuriously correlated variables, thereby injecting noise into dependency modeling. By contrast, TimeFilter achieves the best performance, likely because it dynamically captures variable relationships at the patch level.

Overall, Crisp delivers the best performance across most datasets and prediction horizons, ranking first in the vast majority of both MSE and MAE metrics. These results demonstrate its consistently superior predictive capability. The performance gains are primarily attributable to its selective interaction mechanism, which facilitates more accurate modeling of inter-variable correlations.

### 5.3. Ablation Study (Q2)

In this section, we evaluate the effectiveness of each component of Crisp. We construct the following variants: (1) **w/ MLP** replaces the dual-stream heterogeneous temporal encoder with an MLP layer; (2) **w/o FAS** removes the Frequency-Adaptive Selective Interaction module; (3) **w/o ASM** removes the Adaptive Strict-Blocking Mechanism and uses a standard Transformer instead; (4) **w/o SGF** removes the Spectrum-Gated Feature Filtration module.

As shown in Figure 6, all variants perform worse than Crisp to varying degrees, demonstrating that each proposed component is effective and necessary. The w/ MLP variant exhibits noticeably degraded performance, confirming that the dual-stream temporal encoder is crucial for capturing temporal dynamics. The w/o FAS variant, which essentially adopts a channel-isolation strategy, also yields inferior forecasting accuracy, indicating that selective inter-variable interactions are essential for performance improvement. In addition, the Spectrum-Gated Feature (SGF) Filtration module further refines representations by suppressing noise, contributing to the overall performance gains. In contrast, the complete Crisp model achieves the best performance.

### 5.4. Efficiency Analysis (Q3)

We evaluate the model complexity on the METR-LA dataset with the input-96 / prediction-12 setting, reporting training speed (epochs/s), memory consumption (MB), and MAE.

*Table 2.* Multivariate time series forecasting results. The look-back length $L$ is fixed at 96, and "*" marks an abbreviated model name for clarity. "Avg." refers to the average results across all horizons. The best results are highlighted in **bold**, while the second-best results are underlined.

| Model | | Crisp (Ours) | | TQNet (2025) | | DUET (2025) | | TimeFilter (2025) | | SimpleTM (2025) | | WPMixer (2025) | | xPatch (2025) | | iTrans* (2024) | | MultiPatch* (2024) | | TimesNet (2023) | | PatchTST (2023) | | TimeMixer (2023) | | DLinear (2023) | |
|---|---|---|---|---|---|---|---|---|---|---|---|---|---|---|---|---|---|---|---|---|---|---|---|---|---|---|---|
| | Metric | MSE | MAE | MSE | MAE | MSE | MAE | MSE | MAE | MSE | MAE | MSE | MAE | MSE | MAE | MSE | MAE | MSE | MAE | MSE | MAE | MSE | MAE | MSE | MAE | MSE | MAE |
| ETTh1 | 96 | 0.376 | 0.392 | 0.371 | 0.393 | 0.377 | 0.393 | 0.390 | 0.401 | 0.379 | 0.396 | 0.386 | 0.397 | 0.383 | 0.401 | 0.386 | 0.405 | 0.382 | 0.401 | 0.384 | 0.402 | 0.414 | 0.419 | 0.378 | 0.400 | 0.386 | 0.400 |
| | 192 | 0.430 | 0.425 | 0.428 | 0.426 | 0.429 | 0.425 | 0.445 | 0.435 | 0.438 | 0.431 | 0.436 | 0.431 | 0.431 | 0.427 | 0.441 | 0.436 | 0.435 | 0.437 | 0.436 | 0.429 | 0.460 | 0.445 | 0.432 | 0.429 | 0.437 | 0.432 |
| | 336 | 0.466 | 0.440 | 0.476 | 0.446 | 0.471 | 0.446 | 0.487 | 0.451 | 0.474 | 0.445 | 0.487 | 0.451 | 0.483 | 0.455 | 0.487 | 0.458 | 0.466 | 0.451 | 0.491 | 0.469 | 0.501 | 0.466 | 0.481 | 0.451 | 0.481 | 0.459 |
| | 720 | 0.473 | 0.464 | 0.487 | 0.470 | 0.496 | 0.480 | 0.478 | 0.467 | 0.489 | 0.477 | 0.511 | 0.485 | 0.486 | 0.474 | 0.503 | 0.491 | 0.498 | 0.486 | 0.521 | 0.500 | 0.500 | 0.488 | 0.493 | 0.477 | 0.519 | 0.516 |
| | Avg | 0.436 | 0.430 | 0.441 | 0.434 | 0.443 | 0.436 | 0.450 | 0.439 | 0.445 | 0.437 | 0.455 | 0.441 | 0.446 | 0.439 | 0.454 | 0.448 | 0.445 | 0.444 | 0.458 | 0.450 | 0.469 | 0.455 | 0.446 | 0.439 | 0.456 | 0.452 |
| ETTh2 | 96 | 0.279 | 0.333 | 0.295 | 0.343 | 0.296 | 0.345 | 0.291 | 0.340 | 0.308 | 0.356 | 0.299 | 0.350 | 0.291 | 0.343 | 0.297 | 0.349 | 0.294 | 0.347 | 0.340 | 0.374 | 0.302 | 0.348 | 0.284 | 0.331 | 0.333 | 0.387 |
| | 192 | 0.349 | 0.381 | 0.367 | 0.393 | 0.368 | 0.389 | 0.363 | 0.397 | 0.383 | 0.404 | 0.371 | 0.395 | 0.372 | 0.392 | 0.380 | 0.400 | 0.374 | 0.395 | 0.402 | 0.414 | 0.388 | 0.400 | 0.364 | 0.382 | 0.477 | 0.476 |
| | 336 | 0.403 | 0.420 | 0.417 | 0.427 | 0.411 | 0.422 | 0.408 | 0.427 | 0.422 | 0.436 | 0.426 | 0.433 | 0.423 | 0.433 | 0.428 | 0.432 | 0.418 | 0.431 | 0.452 | 0.452 | 0.426 | 0.433 | 0.406 | 0.416 | 0.594 | 0.541 |
| | 720 | 0.413 | 0.435 | 0.433 | 0.446 | 0.412 | 0.434 | 0.458 | 0.461 | 0.442 | 0.454 | 0.433 | 0.449 | 0.430 | 0.451 | 0.427 | 0.445 | 0.436 | 0.454 | 0.462 | 0.468 | 0.431 | 0.446 | 0.423 | 0.436 | 0.831 | 0.657 |
| | Avg | 0.361 | 0.392 | 0.378 | 0.402 | 0.372 | 0.397 | 0.380 | 0.406 | 0.389 | 0.412 | 0.382 | 0.407 | 0.379 | 0.405 | 0.383 | 0.407 | 0.380 | 0.407 | 0.414 | 0.427 | 0.387 | 0.407 | 0.369 | 0.391 | 0.559 | 0.515 |
| ETTm1 | 96 | 0.308 | 0.343 | 0.311 | 0.353 | 0.324 | 0.354 | 0.319 | 0.357 | 0.323 | 0.364 | 0.329 | 0.364 | 0.332 | 0.369 | 0.334 | 0.368 | 0.315 | 0.355 | 0.338 | 0.375 | 0.329 | 0.367 | 0.315 | 0.343 | 0.345 | 0.372 |
| | 192 | 0.354 | 0.366 | 0.356 | 0.378 | 0.369 | 0.379 | 0.355 | 0.379 | 0.371 | 0.389 | 0.383 | 0.395 | 0.371 | 0.387 | 0.377 | 0.391 | 0.360 | 0.383 | 0.374 | 0.387 | 0.367 | 0.385 | 0.358 | 0.380 | 0.380 | 0.389 |
| | 336 | 0.381 | 0.388 | 0.390 | 0.401 | 0.404 | 0.402 | 0.388 | 0.405 | 0.411 | 0.413 | 0.400 | 0.408 | 0.399 | 0.407 | 0.426 | 0.420 | 0.398 | 0.410 | 0.410 | 0.411 | 0.399 | 0.410 | 0.385 | 0.400 | 0.413 | 0.413 |
| | 720 | 0.448 | 0.428 | 0.452 | 0.440 | 0.463 | 0.437 | 0.448 | 0.437 | 0.460 | 0.445 | 0.487 | 0.453 | 0.463 | 0.443 | 0.491 | 0.459 | 0.467 | 0.447 | 0.478 | 0.450 | 0.454 | 0.439 | 0.455 | 0.443 | 0.474 | 0.453 |
| | Avg | 0.373 | 0.381 | 0.377 | 0.393 | 0.390 | 0.393 | 0.377 | 0.395 | 0.391 | 0.403 | 0.400 | 0.405 | 0.391 | 0.402 | 0.407 | 0.410 | 0.385 | 0.399 | 0.400 | 0.406 | 0.387 | 0.400 | 0.378 | 0.392 | 0.403 | 0.407 |
| ETTm2 | 96 | 0.167 | 0.246 | 0.173 | 0.256 | 0.174 | 0.255 | 0.169 | 0.255 | 0.178 | 0.256 | 0.175 | 0.257 | 0.177 | 0.262 | 0.180 | 0.264 | 0.176 | 0.258 | 0.187 | 0.267 | 0.175 | 0.259 | 0.178 | 0.260 | 0.193 | 0.292 |
| | 192 | 0.230 | 0.290 | 0.238 | 0.298 | 0.243 | 0.302 | 0.235 | 0.299 | 0.242 | 0.300 | 0.238 | 0.297 | 0.244 | 0.306 | 0.250 | 0.309 | 0.241 | 0.304 | 0.249 | 0.309 | 0.241 | 0.302 | 0.246 | 0.306 | 0.284 | 0.362 |
| | 336 | 0.289 | 0.329 | 0.301 | 0.340 | 0.304 | 0.341 | 0.293 | 0.336 | 0.309 | 0.344 | 0.305 | 0.342 | 0.304 | 0.342 | 0.311 | 0.348 | 0.303 | 0.342 | 0.321 | 0.351 | 0.305 | 0.343 | 0.296 | 0.339 | 0.369 | 0.427 |
| | 720 | 0.386 | 0.386 | 0.397 | 0.396 | 0.399 | 0.397 | 0.390 | 0.393 | 0.401 | 0.399 | 0.392 | 0.392 | 0.404 | 0.399 | 0.412 | 0.407 | 0.412 | 0.408 | 0.408 | 0.403 | 0.402 | 0.400 | 0.396 | 0.396 | 0.554 | 0.522 |
| | Avg | 0.268 | 0.313 | 0.277 | 0.323 | 0.280 | 0.324 | 0.272 | 0.321 | 0.282 | 0.325 | 0.277 | 0.322 | 0.282 | 0.327 | 0.288 | 0.332 | 0.283 | 0.328 | 0.291 | 0.333 | 0.281 | 0.326 | 0.279 | 0.325 | 0.350 | 0.401 |
| Electricity | 96 | 0.133 | 0.227 | 0.134 | 0.229 | 0.145 | 0.233 | 0.139 | 0.237 | 0.176 | 0.258 | 0.165 | 0.259 | 0.191 | 0.277 | 0.148 | 0.240 | 0.174 | 0.260 | 0.168 | 0.272 | 0.181 | 0.270 | 0.156 | 0.248 | 0.197 | 0.282 |
| | 192 | 0.152 | 0.242 | 0.154 | 0.247 | 0.163 | 0.248 | 0.158 | 0.253 | 0.184 | 0.267 | 0.177 | 0.266 | 0.193 | 0.278 | 0.162 | 0.253 | 0.182 | 0.267 | 0.184 | 0.289 | 0.188 | 0.274 | 0.170 | 0.259 | 0.196 | 0.285 |
| | 336 | 0.165 | 0.258 | 0.169 | 0.264 | 0.175 | 0.262 | 0.175 | 0.272 | 0.204 | 0.289 | 0.196 | 0.286 | 0.206 | 0.291 | 0.178 | 0.269 | 0.199 | 0.285 | 0.198 | 0.300 | 0.204 | 0.293 | 0.186 | 0.276 | 0.209 | 0.301 |
| | 720 | 0.192 | 0.286 | 0.201 | 0.294 | 0.204 | 0.291 | 0.202 | 0.299 | 0.244 | 0.321 | 0.238 | 0.327 | 0.244 | 0.321 | 0.225 | 0.317 | 0.237 | 0.317 | 0.220 | 0.320 | 0.246 | 0.324 | 0.228 | 0.312 | 0.245 | 0.333 |
| | Avg | 0.161 | 0.253 | 0.165 | 0.259 | 0.172 | 0.259 | 0.169 | 0.265 | 0.202 | 0.284 | 0.194 | 0.284 | 0.208 | 0.292 | 0.178 | 0.270 | 0.198 | 0.282 | 0.193 | 0.295 | 0.205 | 0.290 | 0.185 | 0.274 | 0.212 | 0.300 |
| Solar-Energy | 96 | 0.188 | 0.204 | 0.173 | 0.233 | 0.200 | 0.207 | 0.193 | 0.223 | 0.252 | 0.299 | 0.276 | 0.291 | 0.222 | 0.253 | 0.203 | 0.237 | 0.211 | 0.256 | 0.250 | 0.292 | 0.234 | 0.286 | 0.242 | 0.342 | 0.290 | 0.378 |
| | 192 | 0.218 | 0.227 | 0.199 | 0.257 | 0.228 | 0.233 | 0.226 | 0.249 | 0.306 | 0.332 | 0.298 | 0.342 | 0.244 | 0.276 | 0.233 | 0.261 | 0.245 | 0.284 | 0.296 | 0.318 | 0.267 | 0.310 | 0.285 | 0.380 | 0.320 | 0.398 |
| | 336 | 0.235 | 0.242 | 0.211 | 0.263 | 0.262 | 0.244 | 0.235 | 0.261 | 0.372 | 0.360 | 0.312 | 0.361 | 0.262 | 0.286 | 0.248 | 0.273 | 0.265 | 0.299 | 0.319 | 0.330 | 0.290 | 0.315 | 0.282 | 0.326 | 0.353 | 0.415 |
| | 720 | 0.246 | 0.243 | 0.209 | 0.270 | 0.258 | 0.249 | 0.239 | 0.268 | 0.331 | 0.339 | 0.415 | 0.383 | 0.261 | 0.284 | 0.249 | 0.275 | 0.285 | 0.315 | 0.338 | 0.337 | 0.289 | 0.317 | 0.357 | 0.427 | 0.356 | 0.413 |
| | Avg | 0.222 | 0.229 | 0.198 | 0.256 | 0.237 | 0.233 | 0.223 | 0.250 | 0.315 | 0.333 | 0.325 | 0.344 | 0.247 | 0.275 | 0.233 | 0.262 | 0.252 | 0.288 | 0.301 | 0.319 | 0.270 | 0.307 | 0.291 | 0.369 | 0.330 | 0.401 |
| Weather | 96 | 0.147 | 0.188 | 0.157 | 0.200 | 0.163 | 0.202 | 0.156 | 0.202 | 0.175 | 0.219 | 0.169 | 0.215 | 0.178 | 0.221 | 0.174 | 0.214 | 0.163 | 0.212 | 0.172 | 0.220 | 0.177 | 0.210 | 0.163 | 0.210 | 0.196 | 0.255 |
| | 192 | 0.199 | 0.237 | 0.206 | 0.245 | 0.218 | 0.252 | 0.204 | 0.247 | 0.217 | 0.257 | 0.216 | 0.257 | 0.231 | 0.263 | 0.221 | 0.254 | 0.211 | 0.254 | 0.219 | 0.261 | 0.225 | 0.250 | 0.208 | 0.251 | 0.237 | 0.296 |
| | 336 | 0.258 | 0.282 | 0.262 | 0.287 | 0.274 | 0.294 | 0.261 | 0.290 | 0.282 | 0.302 | 0.272 | 0.298 | 0.283 | 0.300 | 0.278 | 0.296 | 0.273 | 0.299 | 0.280 | 0.306 | 0.278 | 0.290 | 0.268 | 0.295 | 0.283 | 0.335 |
| | 720 | 0.338 | 0.335 | 0.344 | 0.342 | 0.349 | 0.343 | 0.345 | 0.342 | 0.357 | 0.351 | 0.352 | 0.348 | 0.360 | 0.350 | 0.358 | 0.349 | 0.351 | 0.348 | 0.365 | 0.359 | 0.354 | 0.340 | 0.346 | 0.348 | 0.345 | 0.381 |
| | Avg | 0.235 | 0.261 | 0.242 | 0.269 | 0.251 | 0.273 | 0.241 | 0.270 | 0.258 | 0.282 | 0.252 | 0.279 | 0.263 | 0.283 | 0.258 | 0.278 | 0.249 | 0.278 | 0.259 | 0.286 | 0.259 | 0.273 | 0.246 | 0.276 | 0.265 | 0.317 |
| AQWan | 96 | 0.776 | 0.465 | 0.802 | 0.487 | 0.773 | 0.476 | 0.779 | 0.474 | 0.823 | 0.498 | 0.787 | 0.481 | 0.827 | 0.489 | 0.828 | 0.496 | 0.814 | 0.496 | 0.846 | 0.510 | 0.803 | 0.489 | 0.809 | 0.490 | 0.784 | 0.510 |
| | 192 | 0.839 | 0.493 | 0.861 | 0.507 | 0.832 | 0.501 | 0.851 | 0.506 | 0.871 | 0.521 | 0.858 | 0.505 | 0.871 | 0.517 | 0.885 | 0.514 | 0.886 | 0.518 | 0.841 | 0.508 | 0.865 | 0.513 | 0.877 | 0.515 | 0.843 | 0.535 |
| | 336 | 0.850 | 0.500 | 0.880 | 0.518 | 0.852 | 0.515 | 0.872 | 0.514 | 0.932 | 0.587 | 0.880 | 0.517 | 0.893 | 0.511 | 0.912 | 0.5324 | 0.912 | 0.526 | 0.853 | 0.515 | 0.905 | 0.526 | 0.894 | 0.524 | 0.865 | 0.547 |
| | 720 | 0.900 | 0.516 | 0.938 | 0.540 | 0.903 | 0.541 | 0.936 | 0.539 | 0.976 | 0.624 | 0.937 | 0.539 | 0.948 | 0.542 | 0.957 | 0.546 | 0.974 | 0.553 | 0.910 | 0.533 | 0.950 | 0.546 | 0.951 | 0.547 | 0.932 | 0.563 |
| | Avg | 0.841 | 0.493 | 0.870 | 0.513 | 0.840 | 0.508 | 0.859 | 0.508 | 0.900 | 0.557 | 0.865 | 0.511 | 0.885 | 0.515 | 0.895 | 0.522 | 0.897 | 0.523 | 0.863 | 0.516 | 0.881 | 0.518 | 0.883 | 0.519 | 0.856 | 0.539 |
| CzeLan | 96 | 0.180 | 0.230 | 0.193 | 0.241 | 0.200 | 0.259 | 0.185 | 0.236 | 0.221 | 0.261 | 0.193 | 0.242 | 0.214 | 0.263 | 0.217 | 0.262 | 0.276 | 0.311 | 0.206 | 0.268 | 0.203 | 0.252 | 0.229 | 0.276 | 0.352 | 0.422 |
| | 192 | 0.210 | 0.247 | 0.227 | 0.264 | 0.221 | 0.278 | 0.216 | 0.261 | 0.242 | 0.281 | 0.224 | 0.264 | 0.241 | 0.286 | 0.246 | 0.287 | 0.256 | 0.292 | 0.261 | 0.304 | 0.235 | 0.276 | 0.262 | 0.297 | 0.471 | 0.475 |
| | 336 | 0.248 | 0.278 | 0.257 | 0.289 | 0.246 | 0.302 | 0.250 | 0.288 | 0.273 | 0.321 | 0.260 | 0.297 | 0.233 | 0.275 | 0.282 | 0.313 | 0.291 | 0.321 | 0.261 | 0.304 | 0.265 | 0.299 | 0.297 | 0.328 | 0.597 | 0.527 |
| | 720 | 0.318 | 0.329 | 0.321 | 0.341 | 0.298 | 0.333 | 0.327 | 0.331 | 0.361 | 0.373 | 0.306 | 0.327 | 0.351 | 0.343 | 0.341 | 0.354 | 0.350 | 0.358 | 0.330 | 0.356 | 0.324 | 0.340 | 0.358 | 0.371 | 0.803 | 0.604 |
| | Avg | 0.239 | 0.271 | 0.249 | 0.284 | 0.241 | 0.293 | 0.244 | 0.279 | 0.274 | 0.309 | 0.246 | 0.282 | 0.260 | 0.292 | 0.271 | 0.304 | 0.293 | 0.321 | 0.265 | 0.308 | 0.257 | 0.292 | 0.286 | 0.318 | 0.556 | 0.507 |
| ZafNoo | 96 | 0.435 | 0.378 | 0.463 | 0.404 | 0.442 | 0.400 | 0.450 | 0.398 | 0.481 | 0.432 | 0.453 | 0.402 | 0.456 | 0.413 | 0.473 | 0.416 | 0.459 | 0.405 | 0.487 | 0.425 | 0.454 | 0.409 | 0.484 | 0.425 | 0.454 | 0.433 |
| | 192 | 0.519 | 0.429 | 0.539 | 0.446 | 0.515 | 0.443 | 0.521 | 0.439 | 0.530 | 0.488 | 0.532 | 0.442 | 0.526 | 0.447 | 0.558 | 0.461 | 0.528 | 0.444 | 0.557 | 0.469 | 0.525 | 0.448 | 0.567 | 0.470 | 0.521 | 0.465 |
| | 336 | 0.553 | 0.450 | 0.589 | 0.475 | 0.541 | 0.466 | 0.557 | 0.462 | 0.601 | 0.533 | 0.578 | 0.472 | 0.561 | 0.491 | 0.598 | 0.483 | 0.622 | 0.502 | 0.639 | 0.514 | 0.577 | 0.474 | 0.626 | 0.504 | 0.561 | 0.483 |
| | 720 | 0.641 | 0.490 | 0.689 | 0.524 | 0.684 | .524 | 0.647 | 0.506 | 0.739 | 0.557 | 0.651 | 0.505 | 0.660 | 0.512 | 0.684 | 0.524 | 0.718 | 0.543 | 0.724 | 0.550 | 0.661 | 0.515 | 0.728 | 0.552 | 0.640 | 0.506 |
| | Avg | 0.537 | 0.437 | 0.570 | 0.462 | 0.546 | 0.458 | 0.544 | 0.451 | 0.588 | 0.502 | 0.554 | 0.455 | 0.551 | 0.466 | 0.578 | 0.471 | 0.582 | 0.474 | 0.602 | 0.489 | 0.554 | 0.462 | 0.601 | 0.488 | 0.544 | 0.472 |

The results are shown in Figure 5, where we focus on comparisons with several advanced MTS prediction models. Specifically, DLinear features a simple architecture but delivers limited performance. DUET and iTransformer achieve comparable efficiency, as both employ vanilla Transformers to model variable dependencies along the channel dimen-

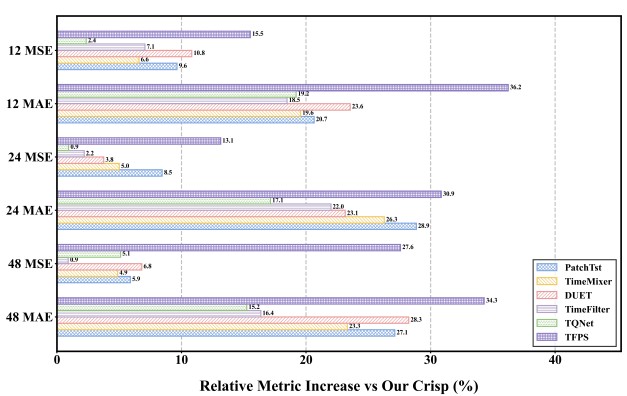

*Figure 4.* Comparison of short-term prediction performance.

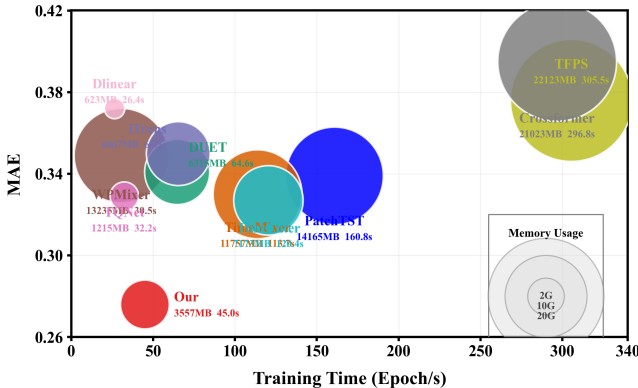

*Figure 5.* Efficiency experiment.

sion. TimeFilter performs more fine-grained patch-level dynamic interactions, which improves modeling capacity but introduces additional computational overhead. In contrast, Crisp achieves high efficiency while simultaneously improving forecasting performance, demonstrating a favorable trade-off between accuracy and computational cost.

### 5.5. Robustness for Missing Values (Q4)

Advanced channel-interaction models typically estimate inter-variable similarities directly from observed time series, making them vulnerable to missing values. As shown in Figure 7, we randomly mask portions of the training data and observe a substantial performance degradation as the missing rate increases. In contrast, our model maintains consistently strong performance under high missing-rate settings, demonstrating superior robustness to incomplete observations.

### 5.6. Evaluation of Key Parameters (Q5)

In Figure 8, we evaluate the sensitivity of the model to the filtering ratio $K$ on the ETTh1 dataset. For ETTh1, the optimal setting $K = 10$ indicates that retaining only 10% of the information is sufficient to achieve strong forecasting performance. In Figure 9, we further visualize the distribution density of the $\alpha$ parameter in $\alpha$-Entmax Transformer,

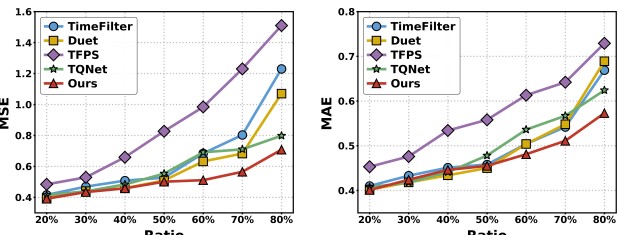

*Figure 7.* Robustness evaluation of interaction strategies.

which is adaptively generated by the hypernetwork based on frequency information to control interaction sparsity. We observe that the mean value of $\alpha$ is 1.83, resulting in a relatively strict truncation behavior. Please note that when $\alpha$ is equal to 1, it degenerates into the ordinary softmax attention.

### 5.7. Case Study (Q6)

We further extract the raw attention weight matrices from ETTh1 for both the vanilla attention mechanism and our proposed attention mechanism, along with the corresponding time series, as shown in Figure 10. The vanilla attention produces relatively diffuse attention patterns. Taking Variable #0 (blue curve) as an example, only Variable #2 exhibits a similar temporal pattern (purple curve), while the remaining variables are weakly related. In contrast, our attention mechanism effectively filters out spurious correlations, en-

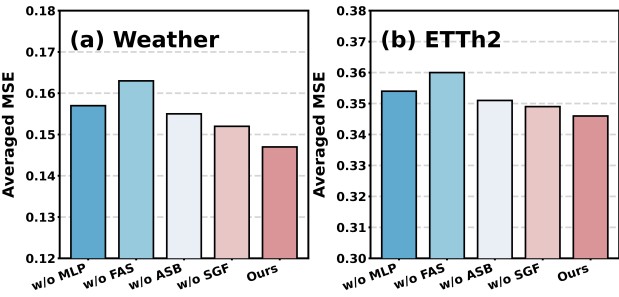

*Figure 6.* Ablation Experiment.

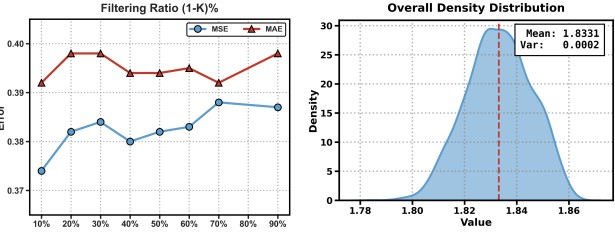

*Figure 8.* Sensitivity.    *Figure 9.* $\alpha$ Distribution.

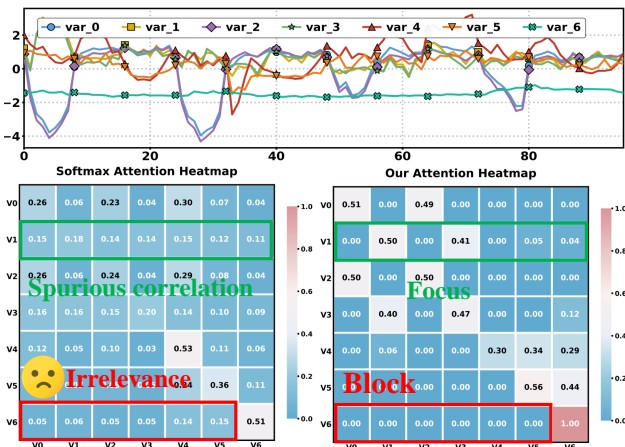

*Figure 10.* Vanilla (left) vs Our (right) attention visualization.

abling more accurate identification of true inter-variable dependencies. This selective interaction leads to more focused attention patterns and ultimately yields more accurate forecasting performance.

## 6. Conclusion

We propose Crisp, which regards spectral consistency as a prerequisite for inter-variable interaction and enforces structured constraints on cross-variable information flow through a dynamic resonance topology. In addition, $\alpha$-Entmax is incorporated to yield strictly sparse attention, zeroing weights for non-resonant neighbors and reducing invalid interactions and their associated gradient interference. Experiments across several real-world datasets verify that Crisp consistently surpasses prior methods, supporting its effectiveness.

## Acknowledgements

This paper is partially supported by the National Natural Science Foundation of China (No.12227901) and the Natural Science Foundation of Jiangsu Province (BK20250482). The AI-driven experiments, simulations and model training were performed on the robotic AI-Scientist platform of Chinese Academy of Science.

## Impact Statement

This work aims to address common real-world time series forecasting problems, thereby advancing the field of time series analysis and providing new methodological perspectives and insights. The proposed approach primarily focuses on improvements at the algorithmic and model levels, without involving sensitive application scenarios or high-stakes decision-making processes. Therefore, it does not introduce new ethical issues beyond those commonly discussed in existing data learning research.

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

# A. Method

## A.1. Algorithm Pseudocode

In Algorithm 1, we present the forward computation process of Crisp. Crisp adopts a heterogeneous architecture that decomposes the time series into trend and seasonal components, and models inter-variable dependencies specifically within the seasonal component.

---

**Algorithm 1:** The pseudocode of Crisp for Multivariate Time Series Forecasting

---

**Input** : Multivariate time series $\mathbf{X} \in \mathbb{R}^{T \times N}$, look-back window $T$, kernel sizes $\mathcal{K}$
**Output** : Prediction $\hat{\mathbf{Y}}$

/* **1.  Normalization & Decomposition** */
1 $\mathbf{X}' \leftarrow \text{RevIN}(\mathbf{X})$ ;                                                    // Standardize input
2 $\hat{\mathbf{X}}, \overline{\mathbf{X}} \leftarrow \text{Decomposition}(\mathbf{X}')$ ;                // Trend & Seasonal split
/* **2.  Trend Branch Processing** */
3 $\hat{\mathbf{Y}}_{trend} \leftarrow \text{Linear}(\hat{\mathbf{X}})$;
/* **3.  Seasonal Branch:  Dual-stream Heterogeneous Encoder** */
4 **for** *each variate* $i \in \{1, \ldots, N\}$ **do**
5    $\mathbf{h}_g^{(i)} \leftarrow \mathbf{W}_g x^{(i)}$ ;                                       // Global trend
6    $\mathbf{h}_1^{(i)} \leftarrow \text{Leaky\_ReLU}(\text{Conv1d}_{1 \times 1}(x^{(i)}))$ ;        // Pointwise
7    $\mathbf{h}_k^{(i)} \leftarrow \text{Pool}(\text{Leaky\_ReLU}(\text{Conv1d}_k(x^{(i)}))), \forall k \in \mathcal{K}$ ;    // Local multi-scale
8    $\mathbf{E}_s^{(i)} \leftarrow \text{Fuse}(\mathbf{h}_g^{(i)}, \mathbf{h}_1^{(i)}, \{\mathbf{h}_k^{(i)}\})$;
9 **end**
10 $\overline{\mathbf{H}} \leftarrow \mathbf{E}_s + \mathbf{E}_t + \mathbf{E}_v$ ;                         // Add temporal & variable priors
/* **4.  Frequency-Adaptive Selective Interaction** */
11 $\mathcal{Z} \leftarrow \text{FFT}(\overline{\mathbf{X}})$;
12 $\mathcal{Z}' \leftarrow \text{LowPassFilter}(\mathcal{Z}, \text{Top-K})$ ;                           // Keep discriminative freqs
13 $\mathbf{E}_f \leftarrow \text{MLP}(\mathcal{Z}')$;
   // Calculate Spectral Coherence Matrix
14 $\mathbf{M}_{i,j} \leftarrow \text{CosSim}(\mathbf{E}_f^{(i)}, \mathbf{E}_f^{(j)}), \quad \forall i, j$;
   // Adaptive Strict-Blocking Mechanism
15 $\mathbf{g} \leftarrow \frac{1}{N} \sum_{i=1}^{N} \mathbf{E}_f^{(i)}$ ;                                 // Global spectral context
16 $\alpha \leftarrow 1 + \sigma(\mathbf{W}_2 \tanh(\mathbf{W}_1 \mathbf{g} + \mathbf{b}_1) + \mathbf{b}_2)$ ;    // Learn sparsity $\alpha \in (1, 2)$
17 $\mathbf{S} \leftarrow \frac{\mathbf{QK}^\top}{\sqrt{D}} + \lambda \cdot \mathbf{M}$ ;                   // Modulate with resonance
18 $\overline{\mathbf{A}} \leftarrow \alpha\text{-Entmax}(\mathbf{S}, \alpha)$ ;                           // Sparse attention weights
19 $\widetilde{\mathbf{H}} \leftarrow \overline{\mathbf{A}}\mathbf{V}$;
   // Spectrum-Gated Filtration
20 $\mathbf{H} \leftarrow \text{GELU}(\mathbf{W}_c \widetilde{\mathbf{H}}) \odot \sigma(\mathbf{W}_g \mathbf{E}_f)$;
21 $\hat{\mathbf{Y}}_{sea} \leftarrow \text{Decoder}(\mathbf{H})$;
/* **5.  Final Prediction** */
22 $\hat{\mathbf{Y}} \leftarrow \text{RevIN}^{-1}(\hat{\mathbf{Y}}_{trend} + \hat{\mathbf{Y}}_{sea})$;
23 **return** $\hat{\mathbf{Y}}$

---

# B. Methodology Supplement

## B.1. $\alpha$-Entmax function

As shown in Figure 11, the figure illustrates the mapping from the *Input Score* $z$ to the output *Probability* $P$ for various values of $\alpha$.

The **Softmax** ($\alpha = 1$) curve asymptotically approaches zero but never reaches it ($P > 0$), resulting in dense outputs.

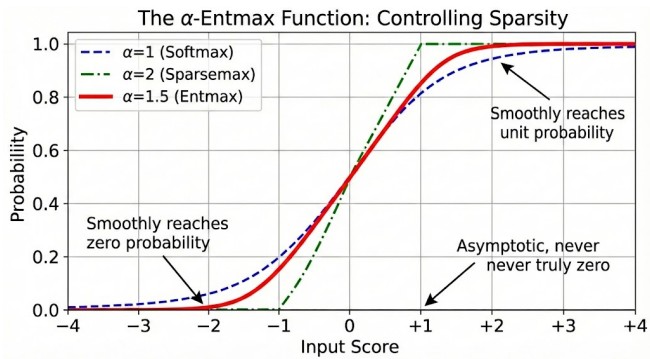

*Figure 11.* $\alpha$-Entmax.

The **Sparsemax** ($\alpha = 2$) function is piecewise linear, creating "hard" sparsity with sharp kinks.

In contrast, the $\alpha$-**Entmax** ($\alpha = 1.5$) curve exhibits *smooth sparsity*: it reaches exactly zero probability with a smooth, differentiable curvature, effectively balancing the properties of Softmax and Sparsemax.

### B.2. Complexity Analysis

Standard dense interaction mechanisms require $\mathcal{O}(N^2)$ memory and computation to generate the attention map. Although Crisp also needs $\mathcal{O}(N^2)$ computation to form the initial similarity matrix, the subsequent weighted aggregation is performed only over a filtered sparse subset. When $\alpha \to 2$, the number of effective interaction neighbors per variable, $k$, satisfies $k \ll N$, reducing the complexity of information flow to $\mathcal{O}(N \cdot k)$.

## C. Experiment

### C.1. Baseline Models

For the baselines, we perform a coarse categorization into four groups: channel-independent models, CNN-based models, MLP-based models, and attention-based models. The latter three categories model interactions using MLP, CNN, and attention mechanisms, respectively.

Since some methods do not explicitly state whether they adopt channel independence, we carefully verify this setting by inspecting their official implementations. In addition, **for models where channel independence is provided only as an optional configuration, we treat them as channel-interactive models, as they are designed to model cross-variable interactions under certain settings**. The detailed groupings are as follows:

- **Channel-independent:** xPatch (Stitsyuk & Choi, 2025), DLinear (Zeng et al., 2023), PatchTST (Nie et al., 2023), CycleNet (Lin et al., 2024), FITS (Xu et al., 2023a), FreTS (Yi et al., 2023), and WPMixer (Murad et al., 2025).

- **CNN-based:** TimesNet (Wu et al., 2023a), MultiPatchFormer (Naghashi et al., 2025), and FilterNet (Yi et al., 2024).

- **Attention-based:** iTransformer (Liu et al., 2024), Crossformer (Zhang & Yan, 2023), CrossGNN (Huang et al., 2023a), Duet (Qiu et al., 2025), TQNet (Lin et al., 2025), TimeFilter (Hu et al., 2025b), and TFPS (Sun et al., 2025).

- **MLP-based:** SOFTS (Han et al., 2024), TimeMixer (Wang et al., 2024c), SimpleTM (Chen et al., 2025), and TimeKAN (Huang et al., 2025).

### C.2. Metrics

We employ mean squared error (MSE) and mean absolute error (MAE) as evaluation metrics, which are formulated as follows:

$$\text{MSE} = \frac{1}{T \times P} \sum_{\tau=1}^{T} \sum_{P=1}^{P} (y_{m,\tau} - \hat{y}_{m,\tau})^2, \tag{15}$$

$$\mathrm{MAE} = \frac{1}{T \times P} \sum_{\tau=1}^{T} \sum_{m=1}^{P} |y_{m,\tau} - \hat{y}_{m,\tau}|. \tag{16}$$

## C.3. Overall Performance

We further compare Crisp with more advanced models, as shown in Table 3. TFPS adopts a dual-domain encoder to capture both time-domain and frequency-domain features, and then applies subspace clustering to dynamically identify different patterns across data segments. However, its performance is relatively poor, possibly due to overfitting caused by excessive model complexity. CrossGNN uses graph neural networks to model dependencies among variables, but dense interactions introduce redundant noise, which degrades performance. SparseTSF and CycleNet emphasize periodic modeling with relatively simple architectures and thus achieve competitive performance. TimeKAN leverages KAN structures to model temporal dynamics and performs well on small datasets such as ETT, but exhibits larger errors on moderately sized datasets like Electricity. In contrast, Crisp consistently maintains strong competitiveness, achieving state-of-the-art performance on most of the evaluated metrics.

*Table 3.* Multivariate time series forecasting results with unified look-back $L = 96$. The best model is shown in **boldface**, and the second-best model is underlined.

| | | Crisp Ours | | TFPS (2025) | | PatchMLP (2025) | | TimeKAN (2025) | | CycleNet (2024) | | SOFTS (2024) | | Crossformer (2024) | | FilterNet (2024) | | FITS (2024) | | FreTS (2023) | | CrossGNN (2023) | |
|---|---|---|---|---|---|---|---|---|---|---|---|---|---|---|---|---|---|---|---|---|---|---|---|
| Method | Metric | MSE | MAE | MSE | MAE | MSE | MAE | MSE | MAE | MSE | MAE | MSE | MAE | MSE | MAE | MSE | MAE | MSE | MAE | MSE | MAE | MSE | MAE |
| ETTm1 | 96 | **0.308** | 0.343 | 0.327 | 0.367 | 0.329 | 0.367 | 0.326 | 0.363 | 0.325 | 0.363 | 0.310 | **0.339** | 0.404 | 0.426 | 0.313 | 0.341 | 0.338 | 0.354 | 0.341 | 0.363 | 0.335 | 0.373 |
| | 192 | **0.354** | **0.366** | 0.374 | 0.395 | 0.377 | 0.394 | 0.359 | 0.384 | 0.366 | 0.382 | 0.367 | 0.369 | 0.450 | 0.451 | 0.369 | 0.368 | 0.385 | 0.377 | 0.386 | 0.387 | 0.372 | 0.390 |
| | 336 | **0.387** | 0.393 | 0.392 | 0.401 | 0.408 | 0.413 | 0.390 | 0.407 | 0.396 | 0.401 | 0.400 | 0.392 | 0.532 | 0.515 | 0.399 | **0.391** | 0.418 | 0.398 | 0.415 | 0.407 | 0.403 | 0.411 |
| | 720 | 0.448 | 0.428 | 0.479 | 0.446 | 0.485 | 0.454 | **0.442** | 0.435 | 0.457 | 0.433 | 0.469 | 0.434 | 0.666 | 0.589 | 0.466 | 0.429 | 0.485 | 0.436 | 0.481 | 0.448 | 0.461 | 0.442 |
| ETTm2 | 96 | 0.167 | **0.246** | 0.170 | 0.255 | 0.178 | 0.262 | 0.177 | 0.259 | **0.166** | 0.248 | 0.172 | 0.249 | 0.310 | 0.331 | 0.171 | 0.250 | 0.183 | 0.259 | 0.184 | 0.272 | 0.176 | 0.266 |
| | 192 | **0.230** | **0.290** | 0.235 | 0.296 | 0.244 | 0.307 | 0.242 | 0.304 | 0.233 | 0.291 | 0.238 | 0.294 | 0.734 | 0.725 | 0.235 | 0.293 | 0.248 | 0.300 | 0.251 | 0.318 | 0.240 | 0.307 |
| | 336 | **0.289** | **0.329** | 0.297 | 0.335 | 0.307 | 0.343 | 0.304 | 0.344 | 0.293 | 0.330 | 0.301 | 0.334 | 0.750 | 0.735 | 0.294 | 0.330 | 0.308 | 0.350 | 0.309 | 0.354 | 0.304 | 0.345 |
| | 720 | **0.386** | **0.386** | 0.401 | 0.397 | 0.420 | 0.413 | 0.400 | 0.401 | 0.395 | 0.389 | 0.403 | 0.394 | 0.769 | 0.765 | 0.393 | 0.389 | 0.409 | 0.417 | 0.417 | 0.420 | 0.406 | 0.400 |
| ETTh1 | 96 | 0.376 | 0.392 | 0.398 | 0.413 | 0.392 | 0.405 | 0.384 | 0.396 | 0.376 | 0.391 | 0.376 | 0.391 | 0.423 | 0.448 | 0.378 | **0.389** | 0.412 | 0.416 | 0.397 | 0.404 | 0.382 | 0.398 |
| | 192 | 0.430 | 0.425 | 0.447 | 0.441 | 0.442 | 0.434 | 0.437 | 0.425 | **0.426** | **0.419** | 0.432 | 0.422 | 0.471 | 0.474 | 0.442 | 0.423 | 0.493 | 0.463 | 0.444 | 0.429 | 0.427 | 0.425 |
| | 336 | 0.466 | 0.440 | 0.484 | 0.461 | 0.483 | 0.455 | 0.476 | **0.439** | 0.464 | 0.439 | 0.470 | 0.440 | 0.570 | 0.546 | 0.490 | 0.446 | 0.493 | 0.463 | 0.487 | 0.453 | 0.465 | 0.445 |
| | 720 | 0.473 | 0.464 | 0.488 | 0.476 | 0.502 | 0.484 | 0.468 | 0.470 | **0.461** | **0.460** | 0.464 | 0.461 | 0.653 | 0.621 | 0.492 | 0.463 | 0.534 | 0.513 | 0.557 | 0.537 | 0.472 | 0.468 |
| ETTh2 | 96 | **0.279** | 0.333 | 0.327 | 0.367 | 0.309 | 0.359 | 0.306 | 0.353 | 0.285 | 0.335 | 0.289 | 0.336 | 0.287 | 0.366 | 0.280 | **0.328** | 0.298 | 0.345 | 0.352 | 0.395 | 0.309 | 0.359 |
| | 192 | **0.349** | **0.381** | 0.374 | 0.395 | 0.405 | 0.419 | 0.375 | 0.392 | 0.373 | 0.391 | 0.366 | 0.386 | 0.414 | 0.492 | 0.362 | **0.381** | 0.388 | 0.400 | 0.411 | 0.428 | 0.390 | 0.406 |
| | 336 | **0.403** | **0.420** | 0.413 | 0.424 | 0.432 | 0.441 | 0.425 | 0.435 | 0.421 | 0.433 | 0.417 | 0.423 | 0.597 | 0.542 | 0.412 | 0.421 | 0.423 | 0.430 | 0.483 | 0.476 | 0.426 | 0.444 |
| | 720 | **0.413** | **0.435** | 0.479 | 0.456 | 0.460 | 0.446 | 0.471 | 0.464 | 0.453 | 0.458 | 0.415 | 0.436 | 1.730 | 1.042 | 0.422 | 0.437 | 0.432 | 0.447 | 0.605 | 0.549 | 0.445 | 0.464 |
| Weather | 96 | **0.147** | **0.188** | 0.154 | 0.202 | 0.165 | 0.211 | 0.163 | 0.209 | 0.170 | 0.216 | 0.165 | 0.198 | 0.162 | 0.208 | 0.155 | 0.193 | 0.173 | 0.213 | 0.174 | 0.212 | 0.159 | 0.218 |
| | 192 | **0.199** | **0.237** | 0.205 | 0.249 | 0.212 | 0.252 | 0.209 | 0.252 | 0.222 | 0.259 | 0.215 | 0.246 | 0.207 | 0.249 | 0.204 | 0.241 | 0.221 | 0.255 | 0.213 | 0.249 | 0.211 | 0.266 |
| | 336 | **0.258** | **0.282** | 0.262 | 0.289 | 0.283 | 0.305 | 0.264 | 0.292 | 0.275 | 0.296 | 0.271 | 0.287 | 0.263 | 0.290 | 0.265 | 0.285 | 0.279 | 0.295 | 0.263 | 0.294 | 0.267 | 0.310 |
| | 720 | **0.338** | **0.335** | 0.344 | 0.342 | 0.357 | 0.352 | 0.340 | 0.343 | 0.349 | 0.345 | 0.351 | 0.339 | **0.338** | 0.340 | 0.354 | 0.342 | 0.359 | 0.342 | **0.338** | 0.352 | 0.352 | 0.362 |
| Electricity | 96 | **0.133** | **0.227** | 0.149 | 0.236 | 0.156 | 0.257 | 0.177 | 0.267 | 0.141 | 0.234 | 0.858 | 0.759 | 0.158 | 0.230 | 0.183 | 0.259 | 0.206 | 0.281 | 0.171 | 0.251 | 0.173 | 0.275 |
| | 192 | **0.152** | **0.242** | 0.162 | 0.253 | 0.174 | 0.270 | 0.182 | 0.272 | 0.155 | 0.247 | 0.859 | 0.759 | 0.206 | 0.277 | 0.189 | 0.267 | 0.203 | 0.281 | 0.181 | 0.262 | 0.195 | 0.288 |
| | 336 | **0.165** | **0.258** | 0.200 | 0.310 | 0.199 | 0.297 | 0.198 | 0.287 | 0.172 | 0.264 | 0.868 | 0.763 | 0.272 | 0.335 | 0.205 | 0.284 | 0.220 | 0.302 | 0.198 | 0.280 | 0.206 | 0.300 |
| | 720 | **0.192** | **0.286** | 0.220 | 0.320 | 0.242 | 0.331 | 0.239 | 0.321 | 0.210 | 0.296 | 0.896 | 0.773 | 0.398 | 0.418 | 0.246 | 0.317 | 0.275 | 0.354 | 0.241 | 0.318 | 0.231 | 0.335 |
| Solar | 96 | **0.188** | **0.204** | 0.211 | 0.218 | 0.218 | 0.245 | 0.260 | 0.301 | 0.209 | 0.260 | 0.234 | 0.231 | 0.219 | 0.314 | 0.228 | 0.238 | 0.374 | 0.338 | 0.255 | 0.264 | 0.283 | 0.353 |
| | 192 | **0.218** | **0.227** | 0.232 | 0.236 | 0.258 | 0.288 | 0.252 | 0.306 | 0.231 | 0.269 | 0.279 | 0.246 | 0.231 | 0.322 | 0.274 | 0.263 | 0.422 | 0.365 | 0.260 | 0.262 | 0.316 | 0.374 |
| | 336 | **0.235** | **0.242** | 0.255 | 0.246 | 0.274 | 0.290 | 0.252 | 0.306 | 0.246 | 0.275 | 0.268 | 0.264 | 0.246 | 0.337 | 0.313 | 0.284 | 0.462 | 0.377 | 0.287 | 0.277 | 0.347 | 0.393 |
| | 720 | **0.246** | **0.243** | 0.257 | 0.250 | 0.269 | 0.296 | 0.252 | 0.306 | 0.255 | 0.274 | 0.277 | 0.258 | 0.280 | 0.363 | 0.306 | 0.279 | 0.453 | 0.363 | 0.307 | 0.282 | 0.348 | 0.389 |

## C.4. Model Compatibility Experiment

In this section, we integrate the proposed core component, Frequency-Adaptive Selective Interaction, into channel-isolation-based models, namely DLinear and PatchTST. We then evaluate their performance on ETTh1 (input-96, prediction-96) for long-term forecasting and METR-LA (input-96, prediction-12) for short-term forecasting. The experimental results are summarized in Table 4. We observe that our technique consistently improves the forecasting performance of these models by enabling a better understanding of inter-variable dependencies.

## C.5. Hyperparameter Sensitivity

We evaluate the impact of two important hyperparameters, the learning rate and batch size, on model performance, as reported in Table 5 and 6. Using ETTh1 (input-96-future-96) as an example, we find that our model is not particularly

*Table 4.* Model compatibility with different backbones.

| Model | DLinear | | DLinear+ours | | PatchTST | | PatchTST+ours | |
|---|---|---|---|---|---|---|---|---|
| | MAE | RMSE | MAE | RMSE | MAE | RMSE | MAE | RMSE |
| ETTh1 | 0.386 | 0.400 | 0.381 | 0.393 | 0.414 | 0.419 | 0.403 | 0.410 |
| METR-LA | 0.461 | 0.349 | 0.450 | 0.389 | 0.466 | 0.330 | 0.458 | 0.327 |

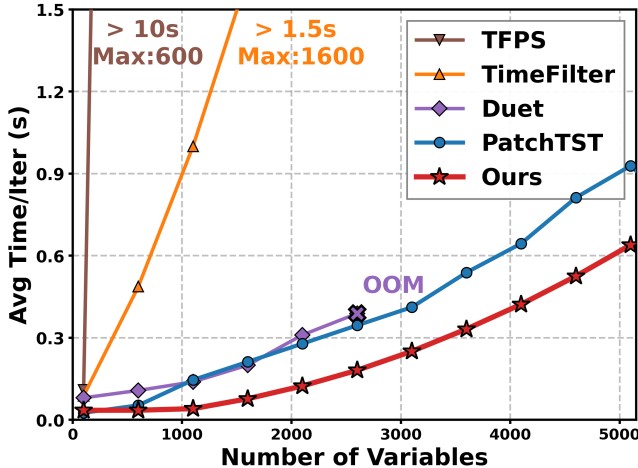

*Figure 12.* Variable expansion experiment.

sensitive to either hyperparameter, demonstrating strong robustness. In general, a larger batch size may increase the risk of overfitting and thus hurt generalization, while an excessively large learning rate can make optimization unstable and lead the model to converge to a suboptimal solution.

*Table 5.* Sensitivity experiment on the learning rate.

| learning_rate | 0.005 | 0.001 | 0.0005 | 0.0001 | 0.0005 | 0.0001 |
|---|---|---|---|---|---|---|
| MAE | 0.413 | 0.388 | 0.383 | 0.376 | 0.382 | 0.381 |
| MSE | 0.417 | 0.396 | 0.393 | 0.392 | 0.396 | 0.397 |

*Table 6.* Sensitivity experiment on the batch size.

| Batch-size | 16 | 32 | 64 | 128 | 256 | 512 |
|---|---|---|---|---|---|---|
| MAE | 0.380 | 0.379 | 0.376 | 0.377 | 0.382 | 0.379 |
| MSE | 0.393 | 0.393 | 0.392 | 0.394 | 0.396 | 0.395 |

### C.6. Model Extensibility Evaluation

In Figure 12, we further compare the scalability of models with respect to the number of variables. We synthesize larger datasets by duplicating the METR-LA dataset, increasing the number of nodes up to 5,100, while setting the batch size to 4 for all models and keeping model hyperparameters fixed to their optimal settings on METR-LA. The experimental results are summarized in the table below. We observe that the two main competitors, TimeFilter and DUET, exhibit a significant increase in computational complexity as the number of nodes grows, indicating their limited scalability for high-dimensional multivariate time series.

## D. Discussion

Although Crisp delivers consistent gains in multivariate dependency modeling, several limitations deserve further discussion. First, our current design mainly adopts discrete Fourier spectra as the spectral prior. An interesting direction is to incorporate richer frequency or time–frequency representations, such as finer spectral statistics or wavelet-based descriptors, to improve interpretability and broaden the types of patterns the model can characterize. This extension is orthogonal to Crisp's central idea and does not affect our main conclusions. Second, the present notion of spectral consistency may be less expressive for cross-variable effects dominated by event-driven behaviors, abrupt propagation, or non-periodic structures. Future work could combine Crisp with causal structural priors or more flexible time-varying dependency modeling, aiming to capture a wider spectrum of dependency types within a unified framework.

