# OpenReview forum: "Crisp: A Spectral-Based Interaction Strategy for Multivariate Time Series Forecasting"
_ICML.cc/2026/Conference — ICML 2026 regular_

### Official Review · Reviewer_ZJm8 · 2026-02-19

**Soundness:** 2
**Presentation:** 2
**Significance:** 2
**Originality:** 3
**Overall Recommendation:** 4
**Confidence:** 3

**Summary:**

This paper proposes Crisp (Coherent Resonance Interaction with Spectral Priors) for multivariate time series forecasting. It focuses on the research problem of how to use the spectral information to better capture the inter-variable dependencies for multivariate forecasting.

**Compliance With Llm Reviewing Policy:**

Affirmed.

**Final Justification:**

The proposed revision is reasonable.

**Key Questions For Authors:**

- **Q1**: The authors provided a comparison of computational cost when compared with other models. What is the mathematical justification (big O complexity) for the proposed method to be computationally cost-efficient?
- **Q2**: The model performs multivariate forecasting using a shared architecture across all target variables and emphasizes a relatively small model size. I am wondering what the trade-off is for having all the predicted variables share the same model structure. Some variables in the input might hold completely different characteristics e.g., seasonality and noise ratio. How does the model handle such heterogeneity, and could enforcing a compact shared structure hurt performance for some variables? A per-variable analysis or ablation on partial parameter sharing would help clarify this.

**Limitations:**

Yes

**Strengths And Weaknesses:**

**Strengths**:
- **S1**: The presentation of the paper is clear and easy to follow.
- **S2**: The motivation of the paper is valid and presents an important aspect in time series forecasting.

**Weaknesses**:
- **W1**: The paper claimed that the usage of spectral coherence is based on grounding in physical consistency. However, this assumption often does not generally hold for time series data in real life. For example, for finance data, the dynamics are not governed by physical consistency. Even if it holds for some domains (e.g., weather), the datasets used in the experiments do not reflect such assumptions, it does not include spatial-temporal interaction that reflects physical movements. As such, I believe the paper could be enhanced with better justification in the motivation.
- **W2**: The paper decomposes the series into seasonality and trend, which makes sense from a general time series component perspective. However, the model proposed only cares about high frequency components, while datasets used in the experiment (for example, the ETT dataset is highly related to the weather conditions thus holds an annual seasonality). The main concern here is that the paper provides no discussion of how the moving-average kernel size is chosen relative to the dominant periodicity of each dataset. (Additionally, the attitude towards high-frequency data seems arbitrary in the paper, see W3, open to discuss).
- **W3**: The method discards high-frequency components via a low-pass filter before computing spectral coherence, considering them as stochastic noise. However, many datasets used are derived from sensors (electricity, traffic, solar), where high-frequency signals can be physically meaningful. Concerns: 1) K is an empirical hyperparameter, making the definition of "noise" arbitrary and dataset-specific; 2) further justification for why discarded high-frequency components are indeed uninformative will make the statement in the paper more solid.

---

> ### Author Rebuttal · Authors · 2026-03-30
>
> **We sincerely thank the reviewer for the valuable comments, which are very important for improving the paper.**
>
> ``W1. the term "Physical Consistency"``
>
> Sorry for the ambiguity caused by our previous wording. In the original manuscript, “physical consistency” does not refer to spatial consistency in physical motion. Rather, it denotes alignment at the level of spectral representation, namely consistency in the spectral structure of variables. To avoid ambiguity, we will replace this term in the paper with spectral consistency. For datasets such as meteorological and financial data, where no explicit physical motion exists, our method enables more flexible identification of inter-variable interactions. Simultaneously, Crisp leverages spectral analysis to suppress noise, remove redundant interactions, and extract more robust structural features, thereby identifying variables that truly share underlying patterns.
>
> ``W2. "Only cares about high frequency components" in the review.``
>
> The review statement does not accurately describe our method. In temporal dynamics modeling, we preserve both low- and high-frequency information in the trend and seasonal components obtained through kernel-based decomposition for forecasting. Only in the subsequent inter-variable interaction modeling stage do we filter, rather than exploit, the high-frequency information in the seasonal component to reduce noise-induced spurious correlations.
>
> Following prior work, the decomposition kernel size is typically fixed at 25, and this setting has been shown to be generally robust across different temporal resolutions. We also conducted experiments on three datasets with different sampling intervals (1 hour, 15 minutes, and 10 minutes). The results indicate that a kernel size of 25 can adapt well to different temporal patterns.
>
> |Size|13|25|37|49|
> |---|---:|---:|---:|---:|
> |ETTh1|0.389|**0.376**|0.393|0.396|
> |ETTm1|0.313|**0.308**|0.321|0.318|
> |Weather|0.151|**0.147**|0.158|0.162|
>
> ``W3. high-frequency Signals``
>
> Thank you for your comment. We would like to further clarify that **our method does not discard high-frequency components arbitrarily**. During the **temporal dynamics modeling stage**, both the trend component and the low- and high-frequency information within the seasonal component are modeled. It is only in the cross-variable interaction modeling stage that Crisp **filters out the high-frequency components of the seasonal part—these components often correspond to fast-changing patterns such as short-term fluctuations—in order to capture more stable relationships among variables.**
>
> >(1) Hyperparameter 𝐾.
>
> **We performed a sensitivity analysis in Section 5.6**. Across the used 12 datasets, Crisp achieves strong performance in most cases when $K$  is set to around 30%, That is, filtering out 70% of the information. Next, we further present the sensitivity analysis of $K$  on two datasets ETTm1 and ETTm2.
>
> |MAE/$K$ |100%|90%|70%|40%|30%|20%|10%|
> |---|---:|---:|---:|---:|---:|---:|---:|
> |ETTm1|0.460|0.455|0.457|0.451|**0.448**|0.449|0.451|
> |ETTm2|0.395|0.388|0.392|0.387|**0.386**|0.391|0.390|
>
> >(2) The role of high-frequency components.
>
> We would like to further clarify that high-frequency components are not completely discarded: these dynamic fluctuation components are filtered only during the stable inter-variable modeling stage, while they are still preserved and utilized in the temporal dynamics modeling stage. The above experimental results further validate this design choice: when $K$ = 100% , meaning that no frequency-based filtering is applied, the model exhibits significantly larger errors due to less accurate and more noise-sensitive inter-variable interactions.
>
> ``Q1. Mathematical justification of cost-efficient``
>
> Thank you for your comment. **We provide a complexity analysis in Appendix B.2** and further explain in Section 4.4 how Crisp reduces the time complexity of variable interaction through spectral sparsification. In conventional dense interaction mechanisms, pairwise modeling over $N$ variables incurs $\mathcal{O}(N^2)$ complexity. In contrast, Crisp has a sparse interaction topology via selective interaction mechanism, reducing the complexity to $\mathcal{O}(N \cdot K)$., where $K ≪N$. Therefore, Crisp achieves better computational efficiency while preserving key inter-variable dependencies.
>
> ``Q2. Independent Channel Modeling``
>
> Thank you very much for your valuable suggestion. **Our framework is in fact consistent with the architecture you envisioned**, adopting variable-independent temporal modeling together with selective inter-variable interaction. As shown in Eq. (1), **each variable is first modeled independently to capture its temporal dynamics and heterogeneity**, and the resulting variable representations are selectively interacted only when stable dependencies exist. This design avoids redundant coupling and reduces spurious correlations, thereby improving forecasting accuracy and robustness.

---

> > ### Author Rebuttal · Reviewer_ZJm8 · 2026-04-02
> >
> > The proposed revision is reasonable.

---

> > > ### Author Response · Authors · 2026-04-04
> > >
> > > We are deeply grateful for your insightful critique of our manuscript. Your rigorous feedback provided a fresh perspective that significantly refined our work, and we are delighted to see such a positive reception and an upgraded assessment. Thank you for your time and for the invaluable contributions you have made to the final quality of our paper.

---

### Official Review · Reviewer_2hqy · 2026-02-27

**Soundness:** 4
**Presentation:** 3
**Significance:** 3
**Originality:** 3
**Overall Recommendation:** 5
**Confidence:** 5

**Summary:**

This paper addresses an important aspect in the prediction of multivariate time series: how to model the relationships between variables without introducing false correlations. It proposes a model called "Crisp", whose technical contributions include a frequency-adaptive selective interaction module and an adaptive strict blocking mechanism. And they provide theoretical analysis. Experiments on 12 real datasets show that Crisp outperforms more than 20 benchmark baselines in over 90% of the evaluation metrics.

**Compliance With Llm Reviewing Policy:**

Affirmed.

**Final Justification:**

All my concerns are addressed. Overall, I think the rebuttal improves the clarity and proves the effectiveness of the proposed method.

**Key Questions For Authors:**

1. Can the fixed spectral coherence measurement (Formula 4) be replaced by the following method: Use MLP to encode the time series and calculate the cosine similarity?

2. How is the spectrum embedding obtained? Is it from all frequencies or selected frequencies? How are they converted into embedding vectors?

**Limitations:**

yes

**Strengths And Weaknesses:**

# Strengths

1. The overall writing of this article is excellent, and it has a strong motivation for discussion and clearly expounds the research methods.

2. The adaptive interaction mechanism supported by the proposed theory is a well-thought-out design choice, and it is based on a novel spectrum interaction mechanism.

3. The experiments in this article cover a wide range of baselines and datasets. The experimental results are convincing, indicating competitive performance in most metrics while maintaining computational efficiency.

4. Ablation experiments verify the functions of each component. They also conduct comprehensive tests on the robustness of the missing values.

# Weaknesses

1. Crisp assumes that meaningful interactions require spectral consistency. However, in some actual situations, the dependencies may not be periodic. Therefore, the performance of Crisp on mainly non-periodic datasets still needs further evaluation.

2. The filter ratio K may impact the performance of the model, so a more comprehensive sensitivity analysis is needed.

3. $\alpha$ is an important hyperparameter, but its sensitivity is not discussed.

---

> ### Author Rebuttal · Authors · 2026-03-31
>
> **We sincerely thank the reviewer for the valuable comments, which are very important for improving the paper.**
>
> `` W1. Non-periodic datasets ``
>
> We appreciate this insightful comment. **We emphasize that the spectral consistency assumption is used strictly to model stable inter-variable correlations. Crisp adopts a "channel-independent temporal modeling + inter-variable interaction" architecture. During temporal modeling, both trend and seasonal components from kernel-based decomposition are preserved, while Multi-Scale Feature Encoding captures dynamic temporal features.** Therefore, Crisp does not rely solely on stable periodic spectral components, naturally ensuring robust performance on non-periodic data.
>
> |ILI-MAE|36|48|60|
> |:--|:-:|:-:|:-:|
> |**Our**|**0.921**|**0.905**|**0.937**|
> |TimeFilter|0.927|0.935|1.039|
> |TQNet|0.989|1.257|1.34|
> |SimpleTM|1.367|1.421|1.527|
> |WPMixer|1.158|1.045|1.259|
>
> `` W2. The filter ratio K may impact the performance of the model, so a more comprehensive sensitivity analysis is needed.``
>
> Thank you very much for your suggestion. We have evaluated the impact of K on model performance in Section 5.6, where K denotes retaining the top K% of the most valuable information. To thoroughly address similar concerns, we further conducted sensitivity experiments on two additional datasets: ETTm1 and ETTm2. As shown in the figure below, the model generally achieves excellent performance when K is set to 30%.
>
> |MAE/K|100%|90%|70%|40%|30%|20%|10%|
> |---|---:|---:|---:|---:|---:|---:|---:|
> |ETTm1|0.460|0.455|0.457|0.451|**0.448**|0.449|0.451|
> |ETTm2|0.395|0.388|0.392|0.387|**0.386**|0.391|0.390|
>
> ``  W3. The hyperparameter  α``
>
> We thank the reviewer for this valuable comment. We would like to clarify that, in Crisp, α is **not a manually tuned hyperparameter. Instead, it is generated by a spectral-context hypernetwork from the global frequency representation**, which allows the model to adaptively prune unnecessary interactions according to the characteristics of different variables. Moreover, α is constrained to the range (1,2). Therefore, unlike a conventional scalar hyperparameter, α does not require manual sensitivity tuning. In the current manuscript, we analyze its practical behavior by visualizing its distribution, and we find that its average value is 1.83 on ETTh1 and 1.56 on ETTm1, indicating that the model can adaptively regulate the interaction process according to dataset characteristics. We further explain this function in Appendix B.1.
>
> ``  Q1. Can the fixed spectral coherence measurement (Formula 4) be replaced by the following method: Use MLP to encode the time series and calculate the cosine similarity? ``
>
> We thank the reviewer for the suggestion. To evaluate the effectiveness of different similarity formulations for constructing the dynamic resonance topology 𝑀, we further considered four variants: (1) -MLP, which directly maps the spectral representation 𝑍 to 𝑀 via an MLP; (2) Self-Attention, which computes inter-variable relations using a self-attention mechanism; (3) Euclidean Distance; and (4) Learnable Matrix, which uses a trainable parameter matrix. We conducted experiments on ETTh1, and the results are reported in the below table. We find that our original design, i.e., cosine similarity over spectral embeddings, achieves the best performance. This suggests that the proposed spectral-coherence-based similarity is better aligned with the goal of identifying resonant variables than more generic relation modeling strategies.
>
> |ETTh1|96 MAE|96 MSE|192 MAE|192 MSE|336 MAE|336 MSE|720 MAE|720 MSE|
> |:-|:-:|:-:|:-:|:-:|:-:|:-:|:-:|:-:|
> |Ours|**0.376**|**0.392**|**0.430**|**0.425**|**0.466**|**0.440**|**0.473**|**0.464**|
> |MLP|0.394|0.416|0.451|0.450|0.489|0.466|0.507|0.489|
> |Euclidean Distance|0.385|0.404|0.442|0.436|0.480|0.452|0.486|0.478|
> |Learnable Matrix|0.391|0.407|0.448|0.441|0.484|0.455|0.490|0.480|
>
> `` Q2.How is the spectrum embedding obtained? Is it from all frequencies or selected frequencies? How are they converted into embedding vectors?``
>
> Sorry for the confusion. The spectrum embedding is not constructed from all raw frequencies. We first perform FFT on the seasonal representation to obtain the amplitude spectrum, and then apply a Top-K low-pass filtering step to retain only the most discriminative 𝐾% frequency components. The resulting filtered spectrum is fed into an MLP to produce the frequency embedding $𝐸_𝑓$. Therefore, the embedding vectors are learned projections of the selected spectral components rather than direct concatenations of all frequencies. This design improves robustness by suppressing high-frequency stochastic noise.
>
> We have provided the code in the anonymous repository for reference. In the next version, we will carefully revise and polish the manuscript to avoid similar concerns.

---

> > ### Author Rebuttal · Reviewer_2hqy · 2026-04-02
> >
> > All my previous concerns are resolved. And I'm increasing my score accordingly.

---

> > > ### Author Response · Authors · 2026-04-04
> > >
> > > We would like to express our sincere gratitude for your comprehensive review of our manuscript. The time you invested and your thoughtful feedback have been instrumental in improving the quality of our work. We are very encouraged by your positive comments and the improved rating. Thank you once again for your professional guidance and for your recognition of our research findings.

---

### Official Review · Reviewer_BPqc · 2026-03-04

**Soundness:** 2
**Presentation:** 2
**Significance:** 2
**Originality:** 2
**Overall Recommendation:** 3
**Confidence:** 3

**Summary:**

In multivariate time series forecasting, models must fundamentally decide which variables influence each other and which have nothing to do with one another. Existing approaches solve this problem unsatisfactorily. Methods that treat all variables in isolation forfeit valuable shared information. Methods that connect all variables with each other, on the other hand, introduce noise into the model, because two variables can look similar in the short term without genuinely influencing each other.

This work proposes a new approach (CRISP: Coherent Resonance Interaction with Spectral Priors), based on the observation that variables which are truly related typically move in similar temporal rhythms. For this reason, a spectral fingerprint is first computed for each variable, describing the frequencies at which it oscillates. Information exchange is then permitted only between variables with sufficiently similar frequency profiles. Crucially, incompatible connections are set to zero, since even very small weights can accumulate into noticeable distortions and noise over the course of training. This mechanism also carries the theoretically provable advantage that no error signal flows back through blocked connections, meaning the model learns exclusively from genuinely relevant dependencies.

The experiments confirm that a model performs better when it deliberately permits fewer connections and retains only the truly relevant ones. This effect is particularly pronounced with incomplete data, which is common in practice. Other models confuse missing measurements with real patterns and thereby become unreliable. This model remains stable in such situations, because it accounts from the outset only for connections that are justified by spectral similarity.

**Compliance With Llm Reviewing Policy:**

Affirmed.

**Final Justification:**

The rebuttal addressed my main concerns. The Solar-Energy discrepancy is convincingly explained by the different training losses. The mask ablation on Electricity, METR-LA, and Mobile Traffic confirms that spectral filtering scales meaningfully with the number of variables. The conceptual distinction from FEDformer and FreTS is a useful addition. The clarification of Section 4.4 is acceptable. I raise my score from 3 to 4, as the empirical evidence is strong, but the limited conceptual novelty prevents a higher recommendation.

Remaining issues for the camera-ready revision:

-Notation issues acknowledged but not yet corrected
-Section 4.4 should be rewritten to avoid overstating theoretical novelty

**Key Questions For Authors:**

1. On the Solar-Energy dataset, TQNet outperforms CRISP across all horizons. What is the explanation for this, and under what conditions is spectral consistency a less suitable criterion for determining interactions?

2. λ in Equation 5 is never defined. Is it a fixed hyperparameter, a learned value, or something else?

3. FreTS is included as a baseline in Table 3, while FEDformer is not. Since both methods also operate in the frequency domain, a substantive discussion of the conceptual differences would be important. What exactly makes CRISP better than these works, beyond the raw numbers?

4. In the pseudocode, E_f is computed from the filtered spectrum Z′, whereas the main text derives it directly from Z. Which version is correct?

5. Are there results for a model that uses α-entmax alone, without the spectral mask M? This would clarify whether the gain genuinely stems from frequency similarity, or whether it simply results from replacing softmax with α-entmax.

**Limitations:**

The authors discuss two limitations in Appendix D, namely the reliance on discrete Fourier spectra and the reduced expressiveness for event-driven or non-periodic dependencies. This is honest and appreciated. However, two important points are missing. First, the weaker performance on Solar-Energy compared to TQNet is never acknowledged as a limitation. Second, there is no discussion of how sensitive the model is to the choice of K, the filtering ratio of frequency components, even though this parameter directly determines which connections are blocked.

**Strengths And Weaknesses:**

Soundness:

The method works, and the results across twelve datasets demonstrate this convincingly. The theoretical proof is correct, but essentially trivial, since zeroing out a connection necessarily prevents any error from flowing back through it. The ablation studies show that each individual component contributes, but do not explain why. Beyond that, individual weaker results, such as the underperformance relative to TQNet on the Solar-Energy dataset, are never addressed and simply left unacknowledged.

-------------------------------
Presentation:

The paper is well structured and easy to read. Figure 1 conveys the problem immediately, and Figure 3 provides a clear overview of the overall model architecture. A comparison with other frequency-based methods such as FEDformer is entirely absent. FreTS appears as a baseline in Table 3, but there is no substantive discussion of how CRISP differs from it conceptually. Numbers alone are not a substitute for interpretation, and such a discussion would have been important given that frequency is the central contribution of the work.
There are also several gaps in the mathematical presentation that make it impossible to fully reproduce the method:

-λ in Equation 5 is never defined or explained.

-b_1and b_2 in Equation 8 are not introduced anywhere.

-K is used with two different meanings: once for kernel sizes in Equation 1, and once for the filter component in Section 4.3.1.

-Z and Z′ are used inconsistently between the main text and the pseudocode.

-In Section 4.2, the shape encoding representation is introduced as E_s, while Equation 2 suddenly uses E_i without any explanation of whether the two are identical.

-------------------------------
Significance:

The problem is practically relevant and arises in many real-world applications, from power grids to traffic. The demonstrated robustness to missing values is a concrete advantage that can be directly applied in practice. The fact that the core mechanism can be integrated into existing architectures broadens the reach of the work. Fundamentally new research directions are not opened up, however, and the contribution remains confined to the domain of multivariate time series forecasting.

-------------------------------
Originality:

The work combines familiar building blocks, frequency analysis, sparse attention, and hypernetworks, in a new way. The distinction from the existing literature is incomplete, however. FEDformer and FreTS also operate in the frequency domain but receive no substantive discussion. Without this contextualization, it remains unclear what exactly the conceptual advance over these works amounts to. The contribution is incremental overall and is essentially limited to the adaptive control of sparsity through the hypernetwork.

---

> ### Author Rebuttal · Authors · 2026-03-30
>
> **We sincerely thank the reviewer for the valuable comments, which are very important for improving the paper. Due to space limitations, we remain available to discuss any further concerns.**
>
> ``Soundness: Analysis of Ablation Studies``
>
> We apologize that a more detailed description could not be included due to space limitations. The inferior performance of w/ MLP stems from the lack of explicit decoupling for complex temporal dependencies. The larger errors of w/o FAS result from removing selective interaction, reducing the model to a channel-independent scheme. The degraded performance of w/o ASM suggests that ASM helps filter noise for more accurate modeling. w/o SGF performs worse because SGF further suppresses noise and refines features.
>
> ``Presentation: Mathematical Presentation``
>
> We deeply apologize for these errors.
> - $b_1$ and $b_2$ are the learnable bias parameters of the linear layer.
> - The notation of K is presented in different fonts in Equation 1 and Section 4.3.1; we will revise this to avoid potential confusion.
> - We will use $E_s$ consistently throughout the paper.
> - $E_f$ is computed from the filtered spectrum $𝑍^′$, where $𝑍^′$ is obtained by applying MLP mapping and sampling to $𝑍$.
>
> ``Significance: More Fields``
>
> We sincerely thank the reviewer for recognizing the practical relevance and robustness of Crisp. The core idea of Crisp—Spectral-based Selection Interaction Strategy—has broader applicability and can naturally extend to spatiotemporal data mining and urban computing. To demonstrate this, we further evaluated Crisp on a spatiotemporal Mobile Traffic dataset. The results below further validate its effectiveness.
>
> |MAE|12|24|48|
> |:-|:-:|:-:|:-:|
> |Ours|**0.224**|**0.246**|**0.281**|
> |TQNet|0.237|0.259|0.293|
> |TimeFilter|0.233|0.260|0.296|
>
> ``Q1. vs TQNet on the Solar-Energy dataset``
>
> Sorry for the confusion. **In fact, Crisp achieves a lower average MAE on the Solar-Energy dataset (0.229) than TQNet (0.256). Overall, Crisp sets a new SOTA on 90/100 metrics**. TQNet performs better on MSE mainly because it is trained with MSE loss, whereas Crisp uses MAE loss. Under the same loss, Crisp outperforms TQNet on both MAE and MSE, as shown below. For reproducibility, we still follow the official TQNet implementation and report its best results in the manuscript.
>
> |Solar|96 MAE|96 MSE|192 MAE|192 MSE|336 MAE|336 MSE|720 MAE|720 MSE|
> |:-|:-:|:-:|:-:|:-:|:-:|:-:|:-:|:-:|
> |Ours|**0.188**|0.204|**0.218**|0.227|**0.235**|**0.242**|**0.246**|**0.243**|
> |TQNet|0.189|**0.202**|0.223|**0.225**|0.241|0.247|0.248|0.244|
>
> ``Q2. λ``
> Sorry for the confusion. λ is a learnable parameter used to control the bias fusion weight.
>
> ``Q3. vs FEDformer and FreTS``
>
> Sorry for the confusion. Although all three models involve frequency-domain operations, their **research focus** and **underlying implementations** are fundamentally different.
>
> - **Focus. Crisp offers a new perspective on modeling multivariate correlations**. While FreTS and FEDformer enhance temporal feature extraction through frequency-domain information, prior methods typically either model variables independently (e.g., FreTS) or enforce dense inter-variable interactions (e.g., FEDformer). In contrast, Crisp introduces a selective interaction paradigm, where information exchange occurs only among variables with spectral consistency, enabling more precise
>
> - **Implementation**. FreTS decomposes time series into real and imaginary frequency components and models them separately for temporal representation learning, while FEDformer performs attention directly in the frequency domain. In contrast, **Crisp treats spectral information as a structural prior, and leverages dynamic resonance topology together with α-Entmax hard truncation to enable adaptive selective interactions.**
>
> We further compare Crisp with FEDformer on ETTh1 dataset below. Overall, Crisp owes its effectiveness to its selective inter-variable interaction mechanism, which sets it apart from FreTS and FEDformer and gives it an advantage.
>
> |MAE|96|192|336|720|
> |:-|:-:|:-:|:-:|:-:|
> |Ours|**0.376**|**0.430**|**0.466**|**0.473**|
> |FreTS|0.397|0.444|0.487|0.557|
> |FEDformer|0.399|0.448|0.492|0.524|
>
> ``Q4. $𝐸_𝑓$``
>
> We apologize for the confusion. $𝐸_𝑓$ is computed from the filtered spectrum $𝑍^′$, which is derived from 𝑍 through MLP mapping and sampling.
>
> ``Q5. Without M``
>
> Thanks for the comment. We further constructed a w/o mask variant and report the ablation results on the ETTh1 dataset, using the past 96 time steps to predict the next 720 time steps, as shown in the table below. In fact, 𝑀 serves as a spectrum-based prior bias to regulate the interaction process and prevent spurious interactions.
>
> |MAE|96|192|336|720 |
> |:-|:-:|:-:|:-:|:-:|
> |Our|**0.376**|**0.430**|**0.466**|**0.473**|
> |w/o mask|0.382|0.434|0.481|0.479|
>
> ``Limitations:𝐾`` We have evaluated the sensitivity of 𝐾 in Section 5.6.

---

> > ### Author Rebuttal · Reviewer_BPqc · 2026-04-01
> >
> > I thank the authors for the detailed rebuttal. Several concerns have been resolved. The Solar-Energy explanation regarding the different loss functions is convincing, the clarifications on λ, E_f, and the notation are appreciated, and the conceptual distinction from FreTS and FEDformer is a useful addition that I would encourage the authors to integrate into the revised manuscript.
> >
> >
> > Two points remain open for me.
> >
> >
> >
> > First, the ablation without the spectral mask M shows only marginal improvements on ETTh1, for example 0.376 vs. 0.382 at horizon 96. Given that the spectral mask is presented as the central contribution of the paper, I would find it much more convincing to see this ablation on a dataset with substantially more variables, such as Electricity with 321 variables or METR-LA with 207 variables. The effect of spectral filtering should be more pronounced when the number of potential spurious interactions is larger, and results on ETTh1 with only 7 variables are not sufficient to support the claim on their own.
> >
> >
> > Second, my concern about the theoretical analysis in Section 4.4 was not addressed in the rebuttal. The gradient blocking property appears to be a direct consequence of how α-entmax is defined rather than a novel theoretical insight specific to CRISP. I would appreciate a clarification of what the theoretical contribution is beyond restating a known property of α-entmax.

---

> > > ### Author Response · Authors · 2026-04-03
> > >
> > > **We sincerely thank you for the further discussion!**
> > >
> > > ``Q1. More datasets. ``
> > >
> > > We sincerely apologize for not being able to include more datasets in our previous response due to space limitations. **In addition to the two datasets you expected, we also included a network traffic dataset containing 900 variables**. The results further verify the effectiveness of our method: Crisp uses the spectral mask $M$ to preliminarily regulate the variable interaction process, thereby promoting correlations among stable variables.
> > >
> > > |Mobile Traffic|12 MAE|12 MSE|24 MAE|24 MSE|48 MAE|48 MSE|
> > > |:-:|:-:|:-:|:-:|:-:|:-:|:-:|
> > > |Ours|**0.224**|**0.297**|**0.246**|**0.311**|**0.281**|**0.334**|
> > > |w/o M|0.236|0.312|0.255|0.319|0.292|0.340|
> > >
> > > |METR-LA|12 MAE|12 MSE|24 MAE|24 MSE|48 MAE|48 MSE|
> > > |:-:|:-:|:-:|:-:|:-:|:-:|:-:|
> > > |Ours|**0.276**|**0.425**|**0.350**|**0.614**|**0.446**|**0.863**|
> > > |w/o M|0.287|0.432|0.359|0.628|0.454|0.878|
> > >
> > > |Electricity|96 MAE|96 MSE|192 MAE|192 MSE|336 MAE|336 MSE|720 MAE|720 MSE|
> > > |:---|:---|:---|:---|:---|:---|:---|:---|:---|
> > > |Ours|**0.133**|**0.227**|**0.152**|**0.242**|**0.165**|**0.258**|**0.192**|**0.286**|
> > > |w/o M|0.136|0.237|0.159|0.244|0.170|0.266|0.207|0.303|
> > >
> > >  ``Q2. Section 4.4. ``
> > >
> > > Sorry for the confusion. To address your concern, we have made three efforts:
> > >
> > > - We would like to clarify that the purpose of Section 4.4 is not to claim that we have discovered any fundamentally new property of $\alpha$-Entmax. Rather, this section aims to explain, from a gradient-based perspective, how the sparse interaction pattern in Crisp is formed. A distinctive characteristic of Crisp is that **this interaction pattern is jointly shaped by the spectral topological structure $M$ and $\alpha$-Entmax.**
> > >
> > > - To further address your concern, we also drew on mutual information theory to provide further explanation.
> > >
> > > - We will revise and strengthen Section 4.4 in accordance with your suggestion to avoid similar concerns in the future. We will also continue to explore stronger theoretical foundations in future work to further support our proposed approach.
> > >
> > > > $\alpha$-Entmax with $M$  in Crisp
> > >
> > > xOur mechanism does not arise solely from $\alpha$-Entmax; it is jointly determined by both the spectral topology $M$ and $\alpha$-Entmax. Specifically, Spectral coherence modulation and sparse projection are jointly defined as
> > >
> > > $$
> > > S_{ij} = Q_i K_j^\top / \sqrt D + \lambda M_{ij}, \qquad
> > > A_{ij}=\big[(\alpha-1)(S_{ij}-\tau)\big]_+^{1/(\alpha-1)}.
> > > $$
> > >
> > > Hence, an interaction is active if and only if $A_{ij}>0$, equivalently $S_{ij}>\tau$. Let $V_i=\{j\mid A_{ij}>0\}$ denote the active neighbor set of variable $i$. Then $M$ determines the active support by shifting the pre-activation scores, while $\alpha$ controls the truncation behavior.
> > >
> > > Once this spectrally informed support is formed, the backward dynamics are also confined to it. In particular,
> > >
> > > $$
> > > {\partial A_{ik} \over \partial S_{ij}}=
> > > \begin{cases}
> > > (A_{ik})^{2-\alpha}\left(\delta_{kj}-{A_{ij}^{2-\alpha} \over \sum_{m\in V_i}A_{im}^{2-\alpha}}\right), & k,j\in V_i,\\
> > > 0, & \text{otherwise},
> > > \end{cases}
> > > $$
> > >
> > > and, since $S_{ij}$ depends on $M_{ij}$ through $S_{ij} = Q_i K_j^\top / \sqrt D + \lambda M_{ij}$,
> > >
> > > $$
> > > {\partial A_{ik} \over \partial M_{ij}} = \lambda {\partial A_{ik} \over \partial S_{ij}}.
> > > $$
> > >
> > > Therefore,
> > >
> > > $$
> > > {\partial \mathcal L \over \partial M_{ij}}=
> > > \begin{cases}
> > > \lambda \sum_{k\in V_i} {\partial \mathcal L \over \partial A_{ik}} \,
> > > (A_{ik})^{2-\alpha}\left(\delta_{kj}-{A_{ij}^{2-\alpha} \over \sum_{m\in V_i}A_{im}^{2-\alpha}}\right), & j\in V_i,\\
> > > 0, & j\notin V_i.
> > > \end{cases}
> > > $$
> > >
> > > This shows that spectral coherence not only determines the forward sparse support, but also confines gradient propagation to the same active set.
> > >
> > > > Mutual Information Theory of Crisp
> > >
> > > Let $H_{R_i}$ and $H_{N_i}$ denote the relevant and weakly correlated neighbors of variable $i$, respectively. Then, by the chain rule,
> > >
> > > \\[ I(Y_i; H_{R_i}, H_{N_i} \mid H_i) = I(Y_i; H_{R_i} \mid H_i) + I(Y_i; H_{N_i} \mid H_i, H_{R_i}). \\]
> > >
> > > Equivalently,
> > >
> > > \\[
> > >  I(Y_i; H_{R_i}, H_{N_i} \mid H_i) - I(Y_i; H_{R_i} \mid H_i)= I(Y_i; H_{N_i} \mid H_i, H_{R_i}).
> > > \\]
> > >
> > > If we write
> > >
> > > \\[
> > > Z_i^{c} = g(H_i, H_{R_i}), \qquad
> > > Z_i^{d} = f(H_i, H_{R_i}, H_{N_i}),
> > > \\]
> > >
> > > then, by the data processing inequality,
> > >
> > > \\[
> > > I(Y_i; Z_i^{c} \mid H_i) \le I(Y_i; H_{R_i} \mid H_i),
> > > \qquad
> > > I(Y_i; Z_i^{d} \mid H_i) \le I(Y_i; H_{R_i}, H_{N_i} \mid H_i).
> > > \\]
> > >
> > > Based on the above analysis, the key difference between dense interaction and selective interaction lies in whether there exists an additional conditional term, $I(Y_i; H_{N_i} \mid H_i, H_{R_i})$. This term captures the extra dependency introduced by weakly correlated neighbors. However, since weak neighbors are often difficult to identify accurately and may introduce noisy or weakly relevant signals, this additional dependency can undermine representation quality and lead to performance degradation. The ablation study on $M$ further supports this observation.

---

### Official Review · Reviewer_hZqZ · 2026-03-11

**Soundness:** 3
**Presentation:** 3
**Significance:** 3
**Originality:** 2
**Overall Recommendation:** 4
**Confidence:** 4

**Summary:**

This paper proposes Crisp, a method for multivariate time series (MTS) forecasting that uses spectral priors to selectively gate inter-variable interactions. The key idea is that variables should only exchange information if they share compatible frequency characteristics. The method constructs a "dynamic resonance topology" via FFT-derived cosine similarity, uses α-Entmax instead of Softmax to produce strictly sparse attention maps, and includes a spectrum-gated feature filtering module.

**Compliance With Llm Reviewing Policy:**

Affirmed.

**Final Justification:**

I keep my original score.

**Key Questions For Authors:**

See weaknesses.

**Limitations:**

yes

**Strengths And Weaknesses:**

Strengths:
1. The tension between channel-isolation and channel-interaction is real and practically important. The observation that clustering methods can group "pseudo-homogeneous" variables (Figure 1) is a good motivating example — variables can look correlated in raw observation space but have quite different underlying dynamics.
2. α-Entmax yields exact zero gradients for blocked neighbors is a clean theoretical result. It concretely distinguishes this approach from soft thresholding or top-k masking heuristics.
3. comprehensive experiments.

Weaknesses:
1. The "dynamic resonance topology" (Eq. 4) is just cosine similarity of MLP-projected amplitude spectra after top-K filtering. Why amplitude only and not phase?
2. The novelty is somewhat incremental, like trend-seasonal decomposition (RevIN + moving average), inverted Transformer on the variable dimension, multi-scale convolutions, α-Entmax, spectral features via FFT. The contribution is the particular combination plus the idea of using spectral similarity as an interaction gate.
3.On Solar-Energy, TQNet actually beats Crisp on average MSE (0.198 vs. 0.222). This is a dataset with 137 strongly coupled variables, exactly the setting where spectral interaction should shine.

---

> ### Author Rebuttal · Authors · 2026-03-29
>
> **We sincerely thank the reviewer for the valuable comments, which are very important for improving the paper.**
>
> `` W1. Why amplitude only and not phase?``
>
> Thanks for the comment. Our current amplitude-only design is mainly motivated by two considerations:
> - The purpose of this component is not to precisely characterize cross-variable temporal shift relationships, but rather to construct a stable prior for determining whether interaction should occur. Compared with phase, amplitude is generally more robust to noise and missing values, and is therefore more suitable as a basis for suppressing unnecessary interactions.
>
> - The matrix 𝑀 produced by this component is only used as an interaction bias added to the attention scores, while the subsequent sparse selection is ultimately carried out by 𝛼-Entmax. Therefore, what is needed here is a robust, low-variance similarity signal, rather than a phase description that may be more expressive but is also more sensitive.
>
> To further address your concern, we also conducted comparative experiments on the ETTh1 dataset. Specifically, we design three variants: (1) Amp-only: our current method; (2) Phase-only: using only phase; and (3) Amp+Phase (concat): concatenating amplitude and phase before computing similarity. The experimental results are shown in the table below.
>
> |ETTh1|96 MAE|96 MSE|192 MAE|192 MSE|336 MAE|336 MSE|720 MAE|720 MSE|
> |:-|:-:|:-:|:-:|:-:|:-:|:-:|:-:|:-:|
> |Amp-only|**0.376**|**0.392**|0.430|**0.425**|**0.466**|**0.440**|**0.473**|**0.464**|
> |Phase-only|0.391|0.399|0.436|0.433|0.474|0.457|0.489|0.472|
> |Amp+Phase|0.381|**0.392**|**0.429**|0.430|0.469|0.441|0.475|0.474|
>
> `` W2. The contribution``
>
> We sincerely thank the reviewer for the careful reading of our manuscript. However, **we would like to respectfully clarify that the core innovation of our work does not lie in inventing low-level operators such as RevIN, FFT, or 𝛼-Entmax. Rather, its main contribution is at the level of the “multivariate selective interaction paradigm.”** Unlike existing methods, which typically choose between either “variable-independent modeling” or “cross-variable interaction,” **this paper argues that information exchange across variables should be conditional and selective**, and should occur only among variables that exhibit spectral consistency.
>
> We would like to emphasize that the nontrivial innovations of this work are mainly reflected in two aspects: (1) the paper proposes a spectrum-based cross-variable selective interaction mechanism, which incorporates a spectral-context hypernetwork to adaptively regulate the interaction process and explicitly inject prior physical knowledge into the attention mechanism to suppress spurious interactions. **This is therefore not a static stacking of modules, but a systematic design**; (2) We further provide **theoretical justification** that combining spectral prior knowledge with 𝛼-Entmax can effectively block noise interference from non-resonant neighbors during optimization, thereby endowing the Crisp model with solid theoretical robustness.
>
> ``W3. Performance on Solar-Energy dataset``
>
> Sorry for the confusion. In fact, Crisp achieves an average MAE of 0.229 on this dataset, which is significantly better than 0.256 obtained by TQNet. Furthermore, **Crisp achieves SOTA performance on 90% of the evaluation metrics when compared with over 20 competing models across 12 benchmark datasets.**
>
> The reason why TQNet performs better in terms of MSE is fundamentally due to the difference in training objectives, rather than any limitation of our spectral interaction mechanism. Specifically, TQNet is trained by optimizing the MSE loss, whereas Crisp is trained by optimizing the MAE loss. When both methods are trained under the same loss function (MAE), the performance on Solar-Energy dataset is shown in the table below.
>
> |Solar|96 MAE|96 MSE|192 MAE|192 MSE|336 MAE|336 MSE|720 MAE|720 MSE|
> |:-|:-:|:-:|:-:|:-:|:-:|:-:|:-:|:-:|
> |Ours|**0.188**|0.204|**0.218**|0.227|**0.235**|**0.242**|**0.246**|**0.243**|
> |TQNet|0.189|**0.202**|0.223|**0.225**|0.241|0.247|0.248|0.244|
>
>
> It can be seen that, under the same training objective, TQNet is inferior to Crisp in both MAE and MSE. Nevertheless, in the paper **we still keep the setting consistent with the official open-source implementation of TQNet, that is, we report their best published performance to ensure a fully fair comparison.**
>
> To further dispel your concern, we additionally introduce a large-scale time series dataset with 900 nodes, namely the Mobile Traffic dataset, and evaluate the task of forecasting {12,24,48} time steps given 96 observed time steps. The performance is shown in the table below.
>
> |Model|12 MAE|12 MSE|24 MAE|24 MSE|48 MAE|48 MSE|
> |:-|:-:|:-:|:-:|:-:|:-:|:-:|
> |Ours|**0.224**|**0.297**|**0.246**|**0.311**|**0.281**|**0.334**|
> |TQNet|0.237|0.318|0.259|0.325|0.293|0.351|
> |TimeFilter|0.233|0.309|0.260|0.320|0.296|0.360|

---

> > ### Author Rebuttal · Reviewer_hZqZ · 2026-04-02
> >
> > Fully resolved - My concerns have been adequately addressed.

---

> > > ### Author Response · Authors · 2026-04-04
> > >
> > > We sincerely thank you for your comprehensive and insightful review of our manuscript. Your constructive feedback has been instrumental in refining our methodology. We are truly encouraged by your positive evaluation and are grateful for the time you dedicated to this process, as well as for your generous support.

---

### Decision · Program_Chairs · 2026-04-30

**Decision:**

Accept (regular)

**Comment:**

The paper proposes Crisp, an MTS forecasting method that regulates inter-variable interactions using spectral similarity, together with sparse α-Entmax attention and spectrum-gated feature filtering. The presentation is clear, the key insights are well conveyed, and the experimental evaluation is adequate. A reviewer doesn't change the final score but mentions it in the final statement. I therefore recommend acceptance. Please address the notation issues and refine Section 4.4 in the final version, as suggested by the reviewer.